# Modeling fibrotic alveolar transitional cells with pluripotent stem cell-derived alveolar organoids

Victoria Ptasinski[1,2,3,4] , Susan J Monkley[5] , Karolina Öst[1], Markus Tammia[5] , Hani N Alsafadi[2,3,4] , Catherine Overed-Sayer[6] , Petra Hazon[1], Darcy E Wagner[2,3,4,*] , Lynne A Murray[6,*]

Repeated injury of the lung epithelium is proposed to be the main driver of idiopathic pulmonary fibrosis (IPF). However, available therapies do not specifically target the epithelium and human models of fibrotic epithelial damage with suitability for drug discovery are lacking. We developed a model of the aberrant epithelial reprogramming observed in IPF using alveolar organoids derived from human-induced pluripotent stem cells stimulated with a cocktail of pro-fibrotic and inflammatory cytokines. Deconvolution of RNA-seq data of alveolar organoids indicated that the fibrosis cocktail rapidly increased the proportion of transitional cell types including the *KRT5*−/*KRT17*+ aberrant basaloid phenotype recently identified in the lungs of IPF patients. We found that epithelial reprogramming and extracellular matrix (ECM) production persisted after removal of the fibrosis cocktail. We evaluated the effect of the two clinically approved compounds for IPF, nintedanib and pirfenidone, and found that they reduced the expression of ECM and pro-fibrotic mediators but did not completely reverse epithelial reprogramming. Thus, our system recapitulates key aspects of IPF and is a promising system for drug discovery.

## Introduction

Idiopathic pulmonary fibrosis (IPF) is a chronic lung disease with a median survival of 2–3 yr (1) characterised by thickening of the alveolar walls and extracellular matrix (ECM) deposition in the distal part of the lung (2). The current approved therapies, nintedanib and pirfenidone, slow down disease progression but are unable to reverse the disease course and thus are not curative (3). Repeated alveolar epithelial injury is proposed to be involved in disease onset and may be a main driver of disease (4). Functional epithelial repair has been shown to be impeded in IPF patients, leading to loss of alveolar epithelial type 1 cells (AEC1) and type 2 cells (AEC2). The lung epithelium in patients with IPF is increasingly recognised as dysregulated and cell populations with transient or indeterminate phenotypes have been shown to appear (5) because of aberrant wound healing responses associated with epithelial reprogramming. This includes the appearance of airway epithelial-like cells in the alveoli (6) and the epithelial-to-mesenchymal transition (EMT) state where epithelial cells acquire mesenchymal markers (7). Recent studies have also identified subpopulations of epithelial cells with increased pro-fibrotic transcriptional activity in the lungs of IPF patients, suggesting that these cells are further driving disease pathogenesis (8, 9, 10, 11, 12). This includes the aberrant basaloid cells, which co-express markers of airway basal cells, mesenchyme, and senescent cells (9, 10), and the transient AEC2-AEC1 state (11, 12). However, the functional contribution of these aberrant subpopulations to IPF pathogenesis has not been studied because it is not presently known what induces this phenotypic change and there is a lack of models that mimic this aspect of epithelial reprogramming in IPF. Despite the urgent need for such models, the access to relevant preclinical systems in which expandable human AEC2 are utilised is limited. Although organoid models can enable maintenance of the AEC2 phenotype in culture (13), the poor proliferation of primary AEC2 after isolation from the lung remains an issue. To address this obstacle, several groups have reported advances in deriving proliferative and functional AEC2 from human embryonic stem cells (ESC) (14) or induced pluripotent stem cells (iPSC) (15, 16, 17, 18). These cells have been used in modeling genetic lung diseases such as Hermansky–Pudlak syndrome-associated interstitial pneumonia (19, 20), complex chronic lung diseases like pulmonary fibrosis (21, 22, 23, 24), and responses to environmental injurious stimuli (25). However, there is still a lack of models focused on fibrotic epithelial reprogramming which are adaptable for drug discovery. Such models should include the use of expandable cells which can be stored as frozen

[1]Bioscience COPD/IPF, Research and Early Development, Respiratory & Immunology, BioPharmaceuticals R&D, AstraZeneca, Gothenburg, Sweden   [2]Department of Experimental Medical Sciences, Lung Bioengineering and Regeneration, Lund University, Lund, Sweden   [3]Wallenberg Centre for Molecular Medicine, Lund University, Lund, Sweden   [4]Lund Stem Cell Center, Lund University, Lund, Sweden   [5]Translational Science and Experimental Medicine, Research and Early Development, Respiratory & Immunology, BioPharmaceuticals R&D, AstraZeneca, Gothenburg, Sweden   [6]Bioscience COPD/IPF, Research and Early Development, Respiratory & Immunology, BioPharmaceuticals R&D, AstraZeneca, Cambridge, UK

Correspondence: lynnemurray@hotmail.com
*Darcy E Wagner and Lynne A Murray are co-senior author

 

banks to allow for high throughput drug screening and cultured in a miniaturized format with stimuli-inducing persistent disease-relevant changes.

In this study, we developed a model of fibrotic alveolar epithelial reprogramming. iPSC-derived AEC2 (iAEC2) from frozen banks which were differentiated according to previously published and well-validated protocols (17, 18, 26) and were subsequently exposed to a cocktail of pro-fibrotic and inflammatory cytokines which has previously been shown to induce early fibrotic changes and alterations in epithelial cell phenotypes in human precision-cut lung slices (PCLS). Similarly, the fibrosis cocktail has been used to evaluate anti-fibrotic compounds in the same system, including their effects on phenotypic markers specific to distal lung epithelium (27, 28, 29). We describe that the stimulation of iAEC2 organoids (termed alveolospheres) with this same fibrosis cocktail induces persistent reprogramming, including cells with a similar transient (also termed aberrant basaloid)-like state as reported in IPF, and production of ECM. We also show that our model is responsive to treatment with nintedanib and pirfenidone, demonstrating that this system can potentially be adapted for drug screening. Thus, this system models several aspects of human IPF allowing for studies of fibrosis induction applicable to drug discovery.

## Results

### Fibrosis cocktail induces transcriptomic changes linked to IPF

To create an expandable and biologically relevant cell source for modeling alveolar epithelial injury in IPF, we differentiated AEC2 from iPSC using previously described and well-characterised protocols (17, 18, 26) (Fig S1A). As the published protocols utilised other iPSC lines, we first validated the efficiency of these protocols in our cell line for deriving AEC2 and their ability to be used as frozen banks. Gating for selection of CD47$^{high}$/CD26$^{low}$ cells enriched for putative lung progenitors expressing NKX-2.1, as previously reported (Fig S1B and C) (26). A 3-d definitive endoderm (DE) induction combined with cell splitting at day 8 of differentiation provided the highest yield of CD47$^{high}$/CD26$^{low}$ cells with the iPSC line used in our study (Fig S1D). Splitting during lung progenitor induction has been suggested to improve the yield in certain cell lines (17). Because we could not see a significant difference between splitting the cells at a ratio of 1:3 and 1:6, we continued with the 1:3 ratio to avoid the sparseness of the cells caused by the 1:6 ratio split (18). Consistent with the findings reported in the originally published protocol for deriving alveolospheres containing lamellar bodies and processed surfactant proteins C and B (SP-C and SP-B) (18), our derived alveolospheres expressed surfactant protein SP-B in intracellular granules and we similarly observed secretion into the lumen of the organoids (Fig S1E). We also evaluated the ability to store the alveolospheres as frozen banks, which allows temporal and logistic control for experiments and potential future application in drug screens. Retention of *SFTPC* reporter expression after cryopreservation of the alveolospheres has been described in the previously published protocols which we based our AEC2 derivations on. These previous studies used

standard cell-freezing conditions containing DMSO (17, 18). We first confirmed that this protocol functions for the alveolospheres derived from our iPSC line. As we did not use *SFTPC* reporter cell lines, we instead assessed bulk levels of the AEC2-characteristic pro-form of surfactant protein C (pro-SP-C) expression to assess whether SP-C-producing cells were still present after cryopreservation. We found that the expression of the AEC2-specific marker pro-SP-C (30) was retained for at least two passages after cryopreservation, thawing, and expansion, at which point the cells reached a suitable number for experiments (Fig S1F). Similarly to what we had observed for SP-B, the alveolospheres expressed pro-SP-C in cytosolic vesicular-like patterns (Fig S2). The presence of such vesicular-like cytosolic patterns, together with an extracellular and secreted pattern of surfactant staining in our alveolospheres is highly indicative of a mature AEC2-like phenotype and is in line with the criteria proposed for defining mature AEC2 phenotypes in stem cell-derived cultures (18, 31, 32). Earlier studies have demonstrated that culture of iPSC-derived lung progenitors under high Wnt signaling conditions, as per the protocol we followed in our study, drives differentiation towards alveolar epithelial phenotypes, whereas low Wnt signaling conditions drives differentiation towards airway epithelial cell phenotypes (33). To determine whether any of our cells had acquired airway epithelial phenotypes during their differentiation, we explored the proportion of organoids expressing the AEC2-characteristic marker pro-SP-C and another marker normally expressed in bronchial epithelial cells in the airways, keratin 5 (KRT5) (Fig S3A and B). Although around 74% of the organoids expressed pro-SP-C, we could not detect any organoids with visible KRT5 expression (Fig S3C). Thus, our data support the fact that the alveolar differentiation protocol and culture conditions enabled us to generate alveolospheres from our iPSC line using the previously published protocols, and that the performance of our iPSC line is similar with respect to differentiation and cryopreservation.

Next, we explored the possibility to induce responses linked to IPF in the alveolospheres. Alveolospheres formed from single iAEC2 over 14 d were stimulated for 72 h with a fibrosis cocktail (FC) containing TGF-$\beta$, TNF-$\alpha$, platelet-derived growth factor AB (PDGF-AB), and lysophosphatidic acid, previously shown to induce phenotypic changes of epithelial cells in an ex vivo model mimicking alterations seen in IPF (27, 28) (Fig 1A). We found that the organoids significantly decreased in size after FC stimulation compared with the control cocktail (CC) (i.e., containing the diluents) (Fig 1B and C). Moreover, most of the FC-treated alveolospheres exhibited a "dense" morphology (Fig 1D) characterised by lack of a visible lumen and a darker colour (white arrow heads in Fig 1B). A minority of the FC-stimulated organoids retained a visible lumen after stimulation, which we termed "normal" morphology because of their resemblance of the CC-stimulated organoids (purple arrow in Fig 1B).

As we could see a distinct change of morphology in alveolospheres after FC stimulation, we were curious to investigate the underlying molecular effects associated with these changes. We first explored whether the FC stimulation was associated with induction of cell damage markers by measuring LDH release over time in culture (Fig S4A). LDH release was non-significantly increased in the medium already after 24 h of stimulation and the levels

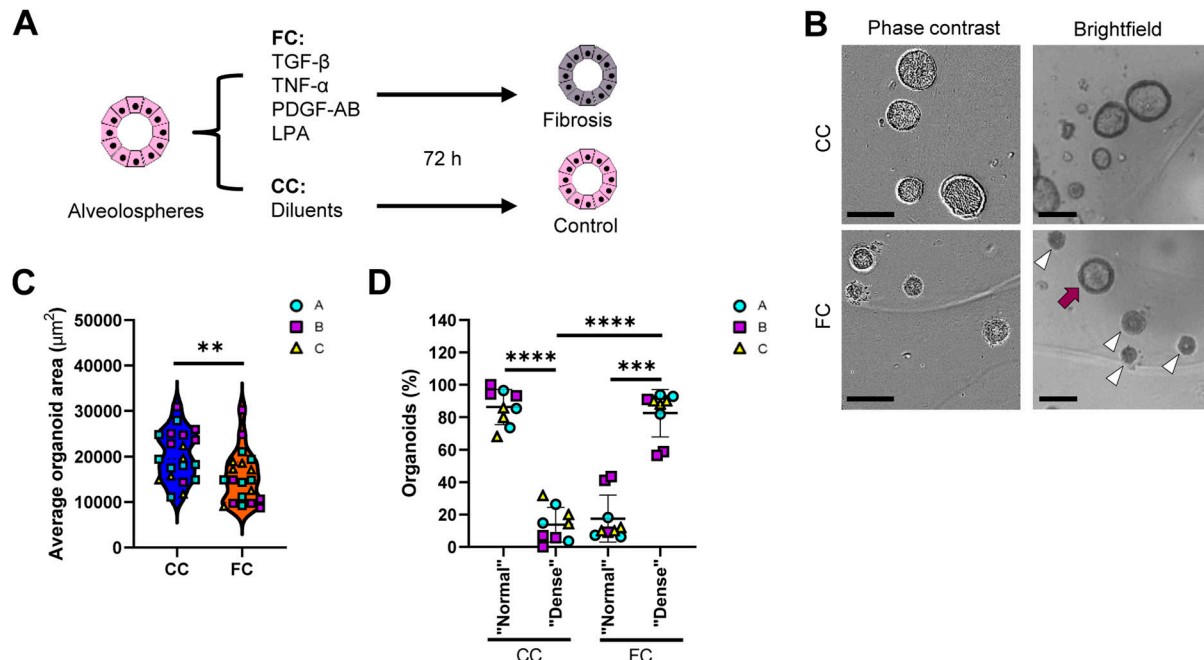

**Figure 1. Fibrosis cocktail induces morphological changes in alveolospheres.**
**(A)** Schematic showing FC stimulation of alveolospheres. **(B)** Representative phase contrast and brightfield images of alveolospheres stimulated with the CC or FC for 72 h. 4x magnification. Purple arrow: organoid with a "normal" morphology, white arrow heads: organoids with "dense" morphology. Scale bar = 250 μm for phase contrast, 200 μm for brightfield. **(C)** Average organoid areas of alveolospheres stimulated with either CC or FC. ** = P < 0.01 by unpaired, two-tailed t-test. n = 5–8 independent images per batch from three batches of alveolospheres. **(D)** Percentage of organoids with a "normal" morphology and "dense" morphology in brightfield images of CC- and FC-stimulated cultures. **** = P < 0.0001, *** = P < 0.001 by repeated measures one-way ANOVA with Sidak's multiple comparisons test. n = 3 independent images per batch from three batches of alveolospheres. See also Figs S1–S4.

remained elevated throughout the stimulation (Fig S4B). The significant increase in LDH release at 72 h of FC stimulation was however not associated with significant loss of organoid number or area of Hoechst staining (Fig S4C and D), suggesting that the FC does not solely induce cell death. To gain insight into the transcriptomic changes induced by the FC, we performed bulk RNA sequencing (RNA-seq) of alveolospheres stimulated with the FC or CC. However, we first explored whether our unstimulated alveolospheres (i.e., CC), contained a high proportion of phenotypic markers associated with cell types other than distal lung epithelial cells (Fig S5A), which could indicate that observed transcriptional changes under FC stimulation were because of the presence of cells which had spontaneously drifted towards cell types not associated with the distal lung, as observed in other studies (34, 35). Importantly, we did not observe high expression of phenotypic markers from other "non-lung" endodermal cell types (e.g., *TFF1* or *CDX2*) or other airway epithelial cell types (e.g., pulmonary neuroendocrine cells) observed in previous studies (34, 35) and only observed meaningful levels of expression for markers associated with distal lung epithelium (Fig S5A) (34, 35). This includes retention of the lung-associated transcription factor *NKX2-1* under FC stimulation (Fig S5B). Although we found *NKX2-1* expression to be significantly reduced (0.7 log$_2$ fold change), this is in line with other public single-cell RNA-seq datasets of human fibrotic primary AEC2 in which the expression of *NKX2-1* is also slightly reduced compared with primary AEC2 isolated from donor control lungs (10). Thus, differential expression in our dataset under FC stimulation is likely

not because of the appearance of "non-lung" cell types in our alveolospheres before FC stimulation or because of spontaneous drift with culture time.

Principal component analysis of our bulk RNA-seq data showed clear separation of samples based on the FC stimulation, explaining 84.7% of the variance (Fig 2A). The top 50 differentially expressed genes by fold change (Fig S6A) included down-regulation of the AEC2 marker *SFTPC* (30) and up-regulation of the fibrosis-associated genes *MMP7* (36) and *SERPINE1* (37) related to ECM remodeling and senescence. Gene enrichment analyses of the differentially expressed genes induced by the FC (SdataF2.1) showed gene ontology enrichment for ECM organization (38), cell adhesion (39), and inflammatory or immune responses (40), all associated with IPF (Fig 2B–D). Pathway analysis demonstrated association with several pathways previously reported to be related to IPF (Fig 2E) including the NF-κB and MAPK signaling pathways (41, 42), and the Hippo/YAP signaling pathway which is altered in epithelial cells in IPF (43). In line with our observations of induced cell death in the alveolospheres with the FC (Fig S4), the pathway analysis predicted simultaneous induction of apoptosis (Figs 2E and S6B) and activation of the p53 signaling pathway related to apoptosis and senescence (44) upon FC stimulation. Notably, we identified up-regulation of a number of genes described in other studies to be associated with the senescence-associated secretory phenotype (45) under FC stimulation (Fig S6C). This is in line with previous work which has shown that epithelial cells express senescence-associated secretory phenotype components both in experimental pulmonary

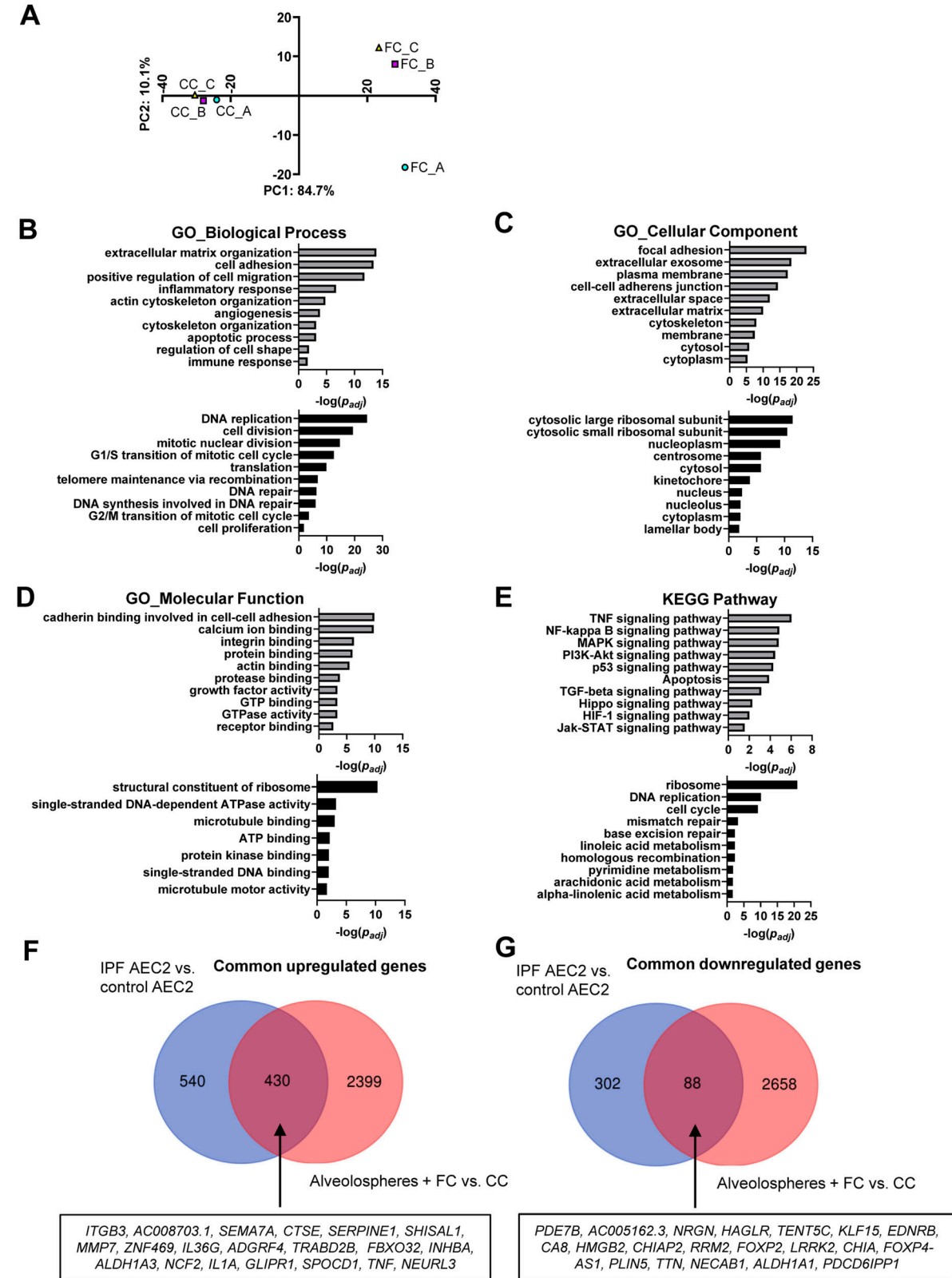

**Figure 2. Fibrosis cocktail induces transcriptomic changes linked to IPF.**
**(A)** PCA plot of RNA-seq data, top 1,000 transcripts by coefficient of variation. *n* = 3 batches of alveolospheres. **(B, C, D, E)** Gene enrichment analyses of the up-regulated (grey) and down-regulated (black) genes according to their gene ontology (GO) biological process, GO cellular component, GO molecular function, and associated Kyoto Encyclopedia of Genes and Genomes pathway. **(F, G)** Venn diagrams of common dysregulated transcripts in FC-stimulated alveolospheres and primary AEC2 isolated from

fibrosis and in IPF tissue and that these are part of disease progression (46). In addition, gene enrichment and pathway analyses of down-regulated genes by the FC showed association with DNA replication, cell division, mitotic nuclear division, cell cycle transition, and cell proliferation (Fig 2B–E). We also found pathways associated with telomere maintenance and DNA repair to be significantly altered under FC conditions; both are known to be impaired in IPF (47) (Figs 2B and S6D). In contrast to the up-regulated genes, the down-regulated genes associated more with intracellular structures such as ribosomes, nuclei, and lamellar bodies, which are important for surfactant processing in AEC2 (32) (Fig 2C). Overall, FC stimulation of the alveolospheres resulted in transcriptional changes which simultaneously recapitulate multiple aspects of IPF (for all gene ontology results, see SdataF2.2–F2.9) and which are not observed in other ex vivo models of IPF (e.g., with TGF-$\beta$ stimulation alone (48)).

Next, we sought to compare changes in gene expression induced by FC in our model with changes seen in distal lung epithelial cells from IPF patients. This was done by comparing our data with a public dataset of primary AEC2 isolated from IPF patients and normal donors (5). FC stimulation of alveolospheres resulted in up-regulation of nearly 50% of the genes significantly up-regulated in AEC2 derived from IPF versus control lung tissue, including genes well-known to be associated with IPF, such as *MMP7* and *SERPINE1* (Fig 2F and SdataF2.10). Among the common down-regulated transcripts (Fig 2G and SdataF2.11), we found genes such as *LRRK2* and *ALDH1A1* which are linked to stemness and AEC2 dysfunction in experimental pulmonary fibrosis (49, 50). Taken together, these results show that the FC-induced transcriptomic changes share similarities to those seen in the distal lung epithelium in IPF.

## Fibrosis cocktail induces epithelial injury

As the loss of alveolar cell phenotypes is a prominent characteristic of IPF, we sought to determine whether the FC alters epithelial phenotypes indicative of epithelial damage in alveolospheres. Consistent with previously published observations in human PCLS (27), we observed down-regulation of the AEC2 markers *SFTPC* and *SFTPB* upon FC stimulation (Fig 3A). The mature, processed form of SP-B (found in normal human lungs at around 6 kD) was decreased in alveolospheres upon FC stimulation (Fig 3B) which indicates loss of the functional AEC2 phenotype (32). We also observed decreases in pro-SP-C expression, which was visible in FC-stimulated organoids with both "normal" and "dense" morphologies (Figs 3C and S7). Therefore, we were curious to see if the loss of the AEC2 phenotype was associated with a phenotypic shift towards acquisition of mesenchymal markers (sometimes termed as EMT) as seen in IPF (7). We observed increased expression of mesenchymal markers with the FC, such as *CDH2* and *VIM* at the gene level (Fig 3A) and on protein level in individual cells (Figs 3C and S7). Importantly, this was not associated with a loss of the epithelial marker *CDH1*/

ECAD indicating that the cells did not lose their epithelial phenotype with FC stimulation (Figs 3A and C and S7).

## Fibrosis cocktail induces epithelial reprogramming into an aberrant basaloid phenotype

As selection of single-cell markers to demonstrate changes in cell phenotype is subjective, we sought to more comprehensively explore changes in cellular phenotypes associated with our alveolosphere model under FC stimulation. For this, we used computational deconvolution of our bulk RNA-seq data using Bisque (51, 52) to predict cellular composition of our alveolospheres with and without FC stimulation. Computational deconvolution uses multiple markers which are obtained from analysing individual cell clusters from single-cell RNA-seq datasets and thus can be viewed as more objective than using single markers to assess changes in cellular phenotype (51). We used a public single-cell RNA-seq dataset (GEO Accession: GSE135893) as a reference filtered to include lung epithelial cells from non-fibrotic controls and IPF donors (10) (Fig 4A). Similar to our previous findings predicted from the use of single, well-established phenotypic markers, deconvolution revealed a decreased proportion of AEC2 in the FC-stimulated alveolospheres in comparison with the CC-stimulated ones (Figs 4B and S8A). Intriguingly, deconvolution predicted an increased proportion of cells with the *KRT5⁻/KRT17⁺* aberrant basaloid phenotype with the FC stimulation (Figs 4B and S8). In line with this, *KRT17* was increased by more than 30 times, whereas *KRT5* was modestly but significantly decreased in our bulk RNA-seq dataset (Fig 4C). Furthermore, we could confirm that KRT17 was significantly increased in the alveolospheres under FC stimulation and that this increase was not linked to a simultaneous increase in KRT5 expression (Figs 4D and E, S3B, and S8B), which supports the induction of the *KRT5⁻/KRT17⁺* aberrant basaloid phenotype (9, 10). The KRT17 expression was most prominent in the FC-stimulated organoids with a "dense" morphology but was also visible in organoids which had retained a "normal" morphology (Fig 4D and F). We also observed FC-induced expression of *KRT8*, a marker which has been reported to be highly expressed in a transient AEC2-AEC1 population in IPF (Fig 4C) (11). Moreover, the *Krt8⁺* alveolar differentiation intermediate cell population in mouse is also known to share transcriptional similarity with the human *KRT5⁻/KRT17⁺*-aberrant basaloid cells (10, 12). Therefore, we performed co-staining of KRT8 and KRT17 and found that KRT8 expression largely coincided with KRT17⁺ cells (12, 53) upon stimulation with the FC (Figs 4D and S8B).

The deconvolution also predicted a modest but consistent induction of cells with the AEC1 phenotype (Fig S8A). The AEC1 phenotype acquired by iPSC-derived cells has in previous studies been defined as cells expressing the markers *PDPN*, *AGER*, and *CAV1* (54). Although we observed significant induction of *CAV1* expression with the FC, we also observed significant decreases of *AGER* and no significant changes of *PDPN* expression (Fig S8C). In addition, we also saw changes in other cell type proportions under FC stimulation such as predicted increases in ciliated cells and *SCGB3A2⁺* cells (Fig S8A). Interestingly, some of the

---

IPF patients (GEO: GSE94555). Selected top 20 dysregulated genes (by absolute fold change) are denoted in the boxes. For (B, C, D, E, F, G): genes selected by *P(adj)* < 0.05, absolute log$_2$ fold change ≥ 0.7. See also Figs S5 and S6, and SdataF2.1–F2.11.
Source data are available for this figure.

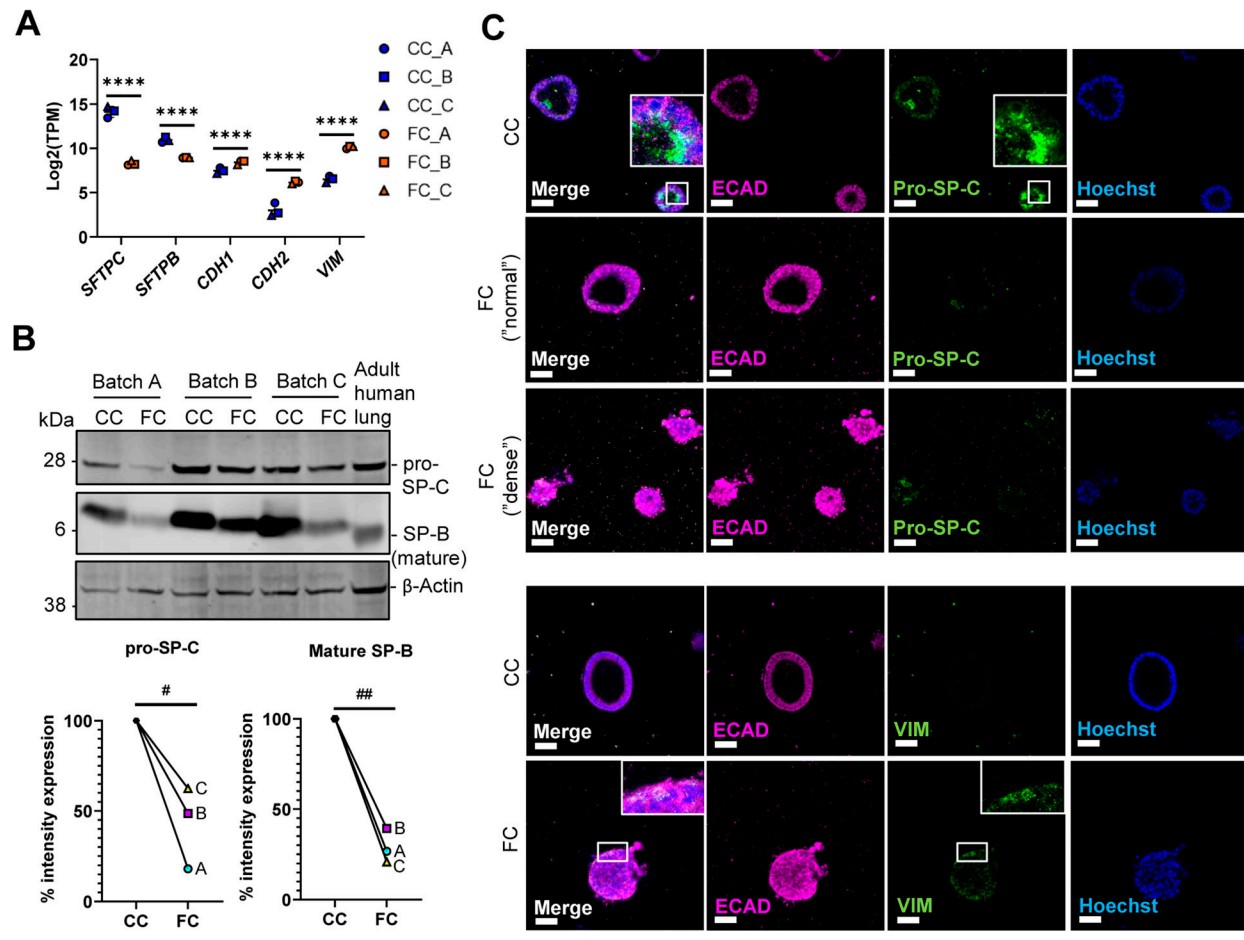

**Figure 3.  Fibrosis cocktail induces epithelial injury.**
**(A)** Expression of the selected epithelial and reprogramming-associated genes. Data presented as means ± SD. **** = P(adj) < 0.0001. n = 3 batches of alveolospheres.
**(B)** Western blot of intracellular protein lysates from alveolospheres stimulated with CC or FC. Adult human lung lysate as positive control. Signal was quantified as band intensity normalised to β-actin and expressed as percentages compared with the respective CC sample set to 100% expression. ## = P < 0.01, # = P < 0.05 by one sample t-test on the percentage intensity compared with a hypothetical value of 100. n = 3 batches of alveolospheres. Blots are cropped from the same membrane and are obtained by sequential blotting as outlined in Supplemental Data 1. **(C)** Immunofluorescence of alveolospheres stimulated with CC or FC. Insert shows a vesicular-like cytosolic pattern of pro-SP-C staining in the alveolospheres (upper panel) or individual cells positive for vimentin (lower panel). Scale bar = 50 μm. See also Fig S7.

classical cell type markers of secretory cells such as *SCGB3A1* and *SCGB3A2* were not elevated in our bulk dataset with FC stimulation (Fig S8C) indicating that other, nontraditional markers which are associated with these cell types in single-cell RNA-seq datasets are driving these changes. This is likely because of the fact that computational deconvolution techniques such as Bisque use a signature of genes rather than individual markers to determine the cell types (51). Alternatively, this may indicate that deconvolution has the potential to capture insights into intermediate states of cellular reprogramming not identified as unique cell clusters in the single-cell dataset used for deconvolution.

## The fibrosis cocktail-induced aberrant reprogramming occurs through alveolar–basal intermediate (ABI) states

As the single-cell sequencing dataset used for the deconvolution in Fig 4 was based on cells obtained from normal and diseased adult

human lung tissue, we next sought to explore cellular composition in a reference single-cell dataset generated from human lung-derived alveolospheres. This is particularly important as recent work has shown that the in vitro culture conditions in which lung epithelial organoids are grown in induced transcriptional phenotypes which are distinct from the cell types they are derived from in vivo (55), potentially indicating that this enables induction of rare transitional cell types which are not possible to isolate in enough numbers from the lung tissue. Thus, we characterised the transcriptional changes induced by FC stimulation in our bulk RNA-seq dataset using a single-cell dataset which characterised changes in primary human AEC2 (hAEC2) over time and co-cultured in the presence of mesenchymal cells derived from adult human lungs (AHLM) (53). Importantly, this reference dataset used pseudotime trajectory analysis to identify the presence of ABI states in their alveolosphere culture (53). Two distinct intermediates were previously identified to exist in organoid culture of hAEC2 cells; ABI1: characterised by retained *SFTPC* levels in cells with acquired *KRT17*

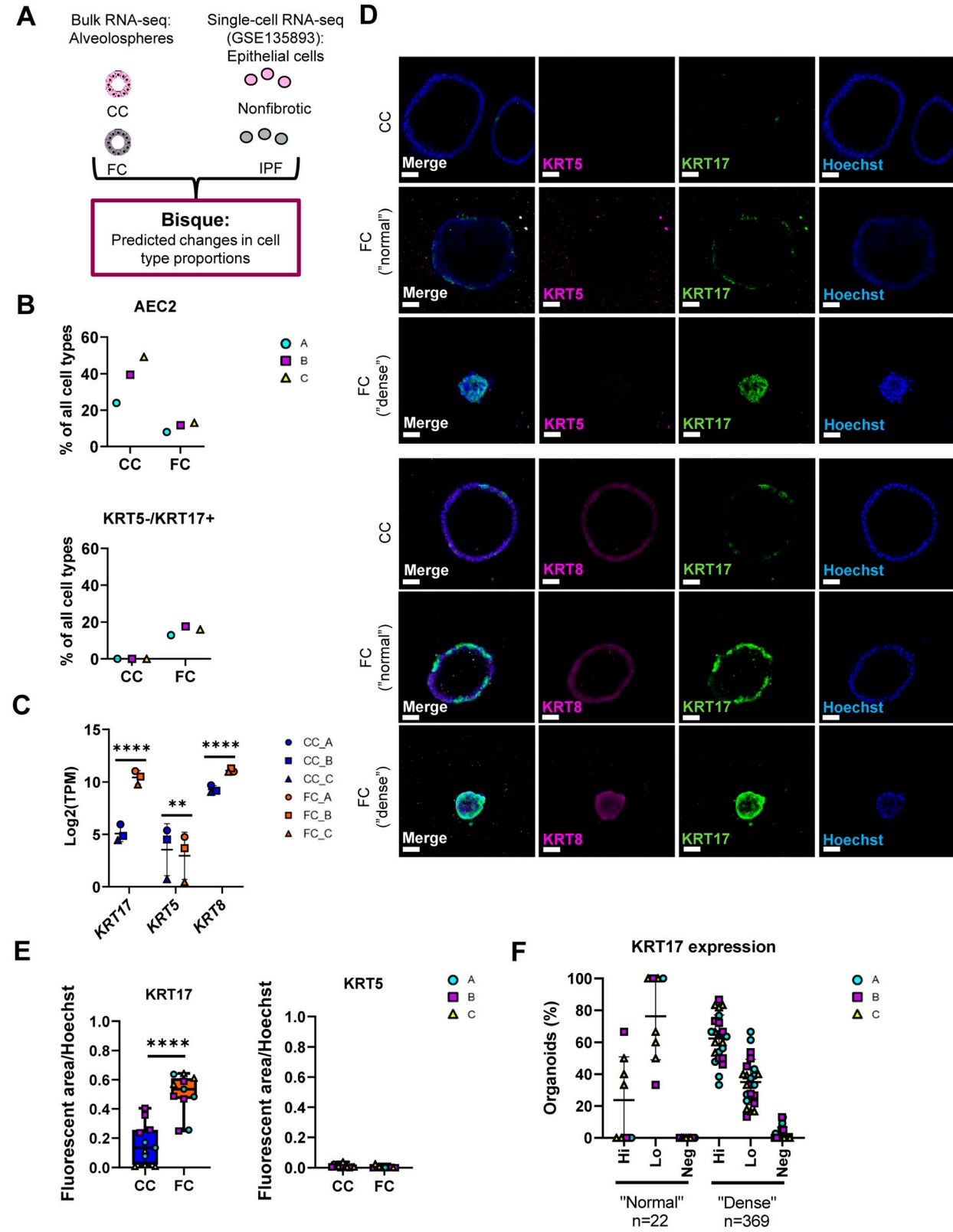

**Figure 4.  Deconvolution analysis predicts increase in aberrant basaloid like-cells by the fibrosis cocktail.**
(A) Schematic depicting Bisque reference-based analysis of epithelial cell populations from non-fibrotic controls and IPF from the publicly available single-cell RNA sequencing dataset (GEO Accession: GSE135893). (B) Selected epithelial cell proportions estimated by Bisque in alveolospheres stimulated with CC or FC. *n* = 3 batches of alveolospheres. (C) Expression of selected epithelial genes in alveolospheres. Data presented as means ± SD. **** = *P(adj)* < 0.0001, ** = *P(adj)* < 0.01. *n* = 3 batches of

expression, and ABI2: defined by cells which express low transcript levels of alveolar markers but high levels of *KRT17*. Both intermediates express high levels of *KRT8*, also associated with the transient AEC2–AEC1 subpopulation and aberrant basaloid cells (11, 12). Therefore, to determine whether the aberrant reprogramming of our alveolospheres induced by the FC follows a similar trajectory as described for primary hAEC2, we performed deconvolution using Bisque with this single-cell dataset (GSE150068) as a reference (Fig 5A) (53). As we were interested in focusing on all potential intermediate and transition states, we first re-clustered and re-annotated the deposited single-cell data (Fig 5B). In addition to identification of all the cell clusters which were previously identified, we identified a transient cluster which was not distinctly observed in the original study, which we named "basal_to_club" because of their co-expression of the basal cell markers *KRT5* and *KRT14* together with low levels of the club cell marker *SCGB1A1* (Figs 5B and S9A) (56). Deconvolution of our bulk RNA-seq dataset using this reference dataset estimated a clear loss of cells with an AEC2 phenotype after FC stimulation (Fig 5C), further confirming this observation across both datasets. However, the use of this dataset allowed us to further explore whether the FC induced potential intermediate cell types during reprogramming which might not be captured in high numbers for lung cell populations sequenced directly after isolation from the lung. Interestingly, we observed predicted increases in the *KRT17*[+] ABI cell proportions, particularly ABI1, indicating that our iAEC2 may follow a similar trajectory of transdifferentiation as described for primary hAEC2 in culture with the lung mesenchyme (53). Moreover, the deconvolution predicted increases in additional airway cell phenotypes such as ciliated cells (Fig 5C), which was another consistent observation across both single-cell datasets.

To further explore how the composition of iAEC2-derived alveolospheres is changed upon stimulation with the FC, we mapped some of the most differentially induced genes by the FC in our dataset to the different clusters (Figs 5D and S9B). The expression of the selected genes, which includes the fibrosis-associated genes *MMP7* (36) and *SERPINE1* (37) and *SEMA7A* and *CTSE*, previously shown to be up-regulated in primary AEC2 isolated from IPF patients (5) (Fig 2F), mostly mapped to the ABI and basal cell clusters (Figs 5D and S9B).

Lastly, we examined FC-induced changes in gene expression in our alveolosphere model for genes identified as unique cell type markers. Cell type marker genes were identified in the hAEC2 organoid single-cell reference dataset based on their differential expression between cell type clusters (Fig 5E). We found a clear loss of expression of the unique gene signature associated with the AEC2 phenotype matching the predicted loss in the AEC2 cell proportion, and acquisition of gene signatures associated with the intermediate states ABI1 and ABI2 and basal cells. For the gene signatures characteristic of the basal to club, club and ciliated cell

clusters, there was no apparent trend of increased or decreased expression of markers indicative of those clusters which indicates that they are not consistently induced by the FC at 72 h. This may be because of the shorter time frame with which we have cultured our cells as these clusters appeared in the original dataset after 21 d in culture. Taken together, our transcriptomic analyses show that the alveolospheres lose their alveolar phenotype after FC stimulation and acquire transcriptomic signatures indicative of aberrant intermediate alveolar–basal states, similar to what is observed over culture time after co-culture of primary hAEC2 organoids with adult human lung mesenchyme (53).

## Fibrosis cocktail induces production of ECM

Previous single-cell studies describing the *KRT5*[−]/*KRT17*[+]-aberrant basaloid cells have also shown that these cells express markers of ECM (9, 10). We therefore assessed the transcriptional changes of matrisome components induced by the FC (Fig 6A) (57). We found that the expression of several matrisome genes was altered by the FC, including up-regulation of *FN1*, *COL1A1*, and *TNC* which are also known to be expressed by aberrant basaloid cells (Fig 6B) (9, 10). We also confirmed that these changes resulted in differences on the protein level and observed increased secretion into the medium of the ECM proteins fibronectin (FN), tenascin C (TNC), and pro-collagen 1α1 (COL1α1) because of FC (Fig 6C–E). As collagen is a major component of the deposited ECM in IPF, we examined the localisation of the collagen 1 (COL1) expression in alveolospheres (Figs 6F and S10). In addition to intracellular staining of the pro-form of COL1, expressed in both CC- and FC-stimulated organoids (Fig S10A), we observed increased secretion of COL1 by the FC-stimulated alveolospheres which was visible on the periphery of the organoids (Figs 6F and S10B), indicating that the cells have a polarised secretion of COL1 on their basal side. In line with our earlier observations, this was not associated with loss of the epithelial marker ECAD (Fig S10A). Taken together, these data show that the FC stimulation is able to induce de novo production and secretion of ECM from epithelial cells.

## Fibrotic responses persist after withdrawal of the fibrosis cocktail

A desired feature of an experimental model of IPF is sustainment of the fibrotic responses over time. Thus, we assessed if the changes we observed over 72 h (3 d) of FC stimulation in our initial experiments related to ECM production, senescence, and aberrant epithelial reprogramming (Fig S11A) persisted after withdrawal of the FC. After stimulation for 3 d, we replaced the FC or CC medium with the normal CK+DCI maintenance medium (Tables S1 and S4) (17, 18) for four additional days (Fig 7A) to see if the changes we induced via FC persisted or more closely resembled levels present in their time- and batch-matched CC control (Fig S11B). We observed

---

alveolospheres. **(D)** Immunofluorescence of alveolospheres stimulated with CC or FC. Scale bar = 50 µm. **(E)** Quantification of the fluorescent area of KRT17 and KRT5 over Hoechst in CC- and FC-stimulated alveolospheres. Boxplots indicate medians with minimum and maximum. *n* = 3–4 independent regions for each condition and each batch of alveolospheres, in total 3 batches. **** = *P* < 0.0001 by unpaired, two-tailed *t*-test. **(F)** Proportion of organoids with "normal" and "dense" morphologies expressing KRT17 in high levels (Hi), low levels (Lo) or no expression (Neg) upon FC stimulation. *n* = 8 independent images for each batch of alveolospheres, in a total of three batches. See also Fig S8.

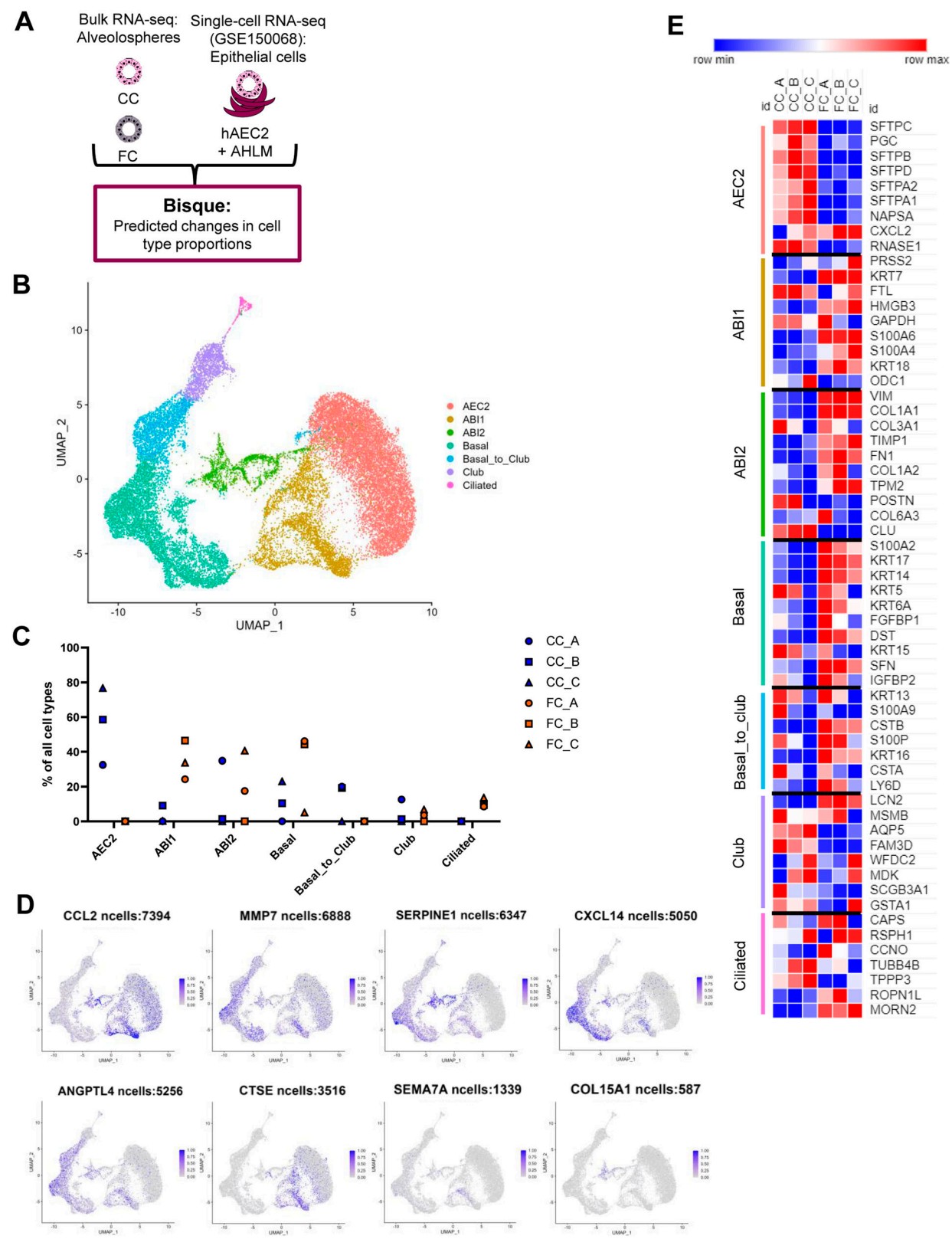

**Figure 5. Deconvolution analysis predicts a trajectory of aberrant reprogramming through alveolar–basal intermediate states.**
**(A)** Schematic depicting Bisque reference-based analysis of epithelial cell clusters from primary human AEC2 organoid co-cultures with adult human lung mesenchyme in the publicly available single-cell RNA sequencing dataset (GEO Accession: GSE150068). **(B)** UMAP plot of the identified cell clusters after re-annotating the deposited single-cell RNA-seq dataset GSE150068. **(C)** Cell proportions estimated by BisqueRNA in alveolospheres stimulated with CC or FC. *n* = 3 batches of alveolospheres.

no reversion of the organoid sizes or change in proportion of "dense" morphology after 4 d of FC withdrawal (Fig 7B–D). As observed earlier, these morphological effects were not associated with a significantly lower organoid count in the cultures (Fig S11C).

We then explored transcriptional changes for selected markers that we had previously observed to be significantly or near significantly (P < 0.07) changed at day 3 of FC stimulation (Fig S11A). We first assessed if ECM gene expression was sustained after the FC withdrawal and found that withdrawal of the FC did not restore the gene expression or protein secretion in the medium of any of the ECM markers FN1, TNC, and COL1A1 to their time- and batch-matched CC controls (Fig 7E and F). Of note, induction and retention of these mesenchymal markers after FC stimulation is not associated with a simultaneous loss of the lung fate-associated transcription factor NKX2-1 (34) nor the epithelial cell marker CDH1 (Fig S12), indicating that the FC-associated expression changes are not linked to a drift of the alveolospheres towards "non-lung" cell phenotypes (34, 35).

Next, we assessed whether the alveolospheres showed signs of recovery towards normal alveolar epithelial states after the FC withdrawal. We also assessed the genes linked with cell cycle arrest, which are often associated with senescence (CDKN1A and CDKN2A), and found no restoration of the expression to their time- and batch-matched CC controls after the FC withdrawal (Fig 7G), indicating persistent cell cycle arrest of the cells. We further explored the expression of genes associated with the alveolar epithelial phenotype and reprogramming (Fig 7H). Although we could not see a significant increase of the surfactant protein SFTPC after the FC withdrawal, we did observe a significant increase in the expression of SFTPB (Fig 7H) indicating partial restoration of the alveolar phenotype. Intriguingly, the aberrant basaloid cell-associated markers KRT8 and KRT17 (Fig 7H and I) (9, 10, 12) were still elevated in the FC-stimulated alveolospheres after withdrawal, suggesting that this aberrant reprogramming was not spontaneously reversed within the assessed time frame. Taken together, this shows that stimulation with the FC induces sustained pro-fibrotic responses, which may allow for exploration of mechanisms which sustain fibrotic epithelial injury and the reprogrammed state after injury.

### FC stimulation of alveolospheres is an adaptable system for drug discovery

Next, we sought to assess if our model has potential for adaptation to drug discovery. We were interested in testing a treatment approach in which we could potentially aim to preserve the normal AEC2 phenotype as opposed to reversing existing and potentially well-established changes in the epithelium, which may be more challenging. We therefore performed prophylactic treatment of alveolospheres with the anti-fibrotic drugs nintedanib and pirfenidone in combination with the FC stimulation. Although pirfenidone and nintedanib are mainly known to target fibroblasts, they

have been previously shown to have differential effects on epithelial cells in pro-fibrotic conditions (28). Therefore, we first performed a dose-response study to assay for compound-mediated LDH cytotoxicity (Fig S13A). Cytotoxicity was not significantly induced after addition of the compounds in concentrations up to 10 µM, although a trend of increased LDH release was seen after addition of 10 µM pirfenidone to the FC (Fig S13B). Thus, higher concentrations of the compounds were not used. We also did not see any significant loss of organoids after treatment with the highest concentrations of the compounds (Fig S13C).

We initially sought to determine which markers were suitable for evaluation of the treatments by looking into the significant changes induced by the FC with the compound vehicle (DMSO) at day 3 (Fig S14A). In line with our earlier observations, we saw significant effects of the FC on markers associated with ECM production (FN1 and TNC) (Fig S14A and B), pro-fibrotic signaling (TGFB1, MMP7 and IGFBP3), cell cycle arrest (CDKN2A), and epithelial reprogramming (SFTPC, SFTPB, KRT17, VIM, and CDH2) at day 3 (Fig S14A). Because other in vitro studies have reported effects on the expression of the ECM marker FN1 and the surfactant protein SFTPC in murine alveolar epithelial cells grown in 2D tissue culture plastic using nintedanib at 1 µM (28), we initially assessed the responses to concentrations of 0.1 and 1 µM on these markers in our system (Fig S15A). Unlike previously published observations, there was no significant effect of the compounds on either of these markers. We therefore focused on evaluating the effects of the compounds at concentrations of 10 µM (Fig 8A).

Intriguingly, treatment of the alveolospheres with 10 µM nintedanib significantly prevented the induction of the "dense" morphology by the FC, which was not seen with pirfenidone (Fig 8B and D). The morphological effect of nintedanib was associated with partially increased organoid sizes, although nonsignificant (Fig 8C). To further determine the molecular changes behind these morphological effects, we evaluated if there were effects of the treatment on the FC-induced production of ECM and pro-fibrotic mediators (Fig 8E–G). Treatment with 10 µM of nintedanib significantly decreased gene expression of FN1 (Fig 8E), and a similar trend was observed for the protein secretion in the medium (P of ANOVA = 0.099) (Fig 8F). In contrast to the effects described in published studies using PCLS from bleomycin-injured mice (28), we did not observe any decrease of COL1A1 gene expression in the alveolospheres after any of the treatments (Fig S15B). However, we did observe significant effects of nintedanib and pirfenidone on the pro-fibrotic mediators TGFB1 and IGFBP3 (Fig 8G) and a trend towards a decrease of MMP7 expression (Fig S15B).

Because the expression of markers associated with alveolar epithelial reprogramming and senescence were sustained after withdrawal of the FC, we investigated if the therapies could prevent their alteration (Figs 8H–J and S15B). Interestingly, treatment with nintedanib significantly reduced the expression of the EMT-associated markers VIM and CDH2, whereas pirfenidone did not (Fig 8H). We also observed trends towards decreased expression of the markers CDKN1A and CDKN2A associated with cell cycle arrest

---

(D) Feature plots showing expression (scaled from 0 to 1) of selected up-regulated genes by FC stimulation. Genes were identified as described in the GitHub repository at: https://github.com/Lung-bioengineering-regeneration-lab/FC_alveolospheres. (E) Heatmap showing expression in alveolospheres of the most highly differentially expressed unique genes from each of the identified cell clusters. Genes selected by P(adj) < 0.05, absolute log₂ fold change ≥ 0.7. See also Fig S9.

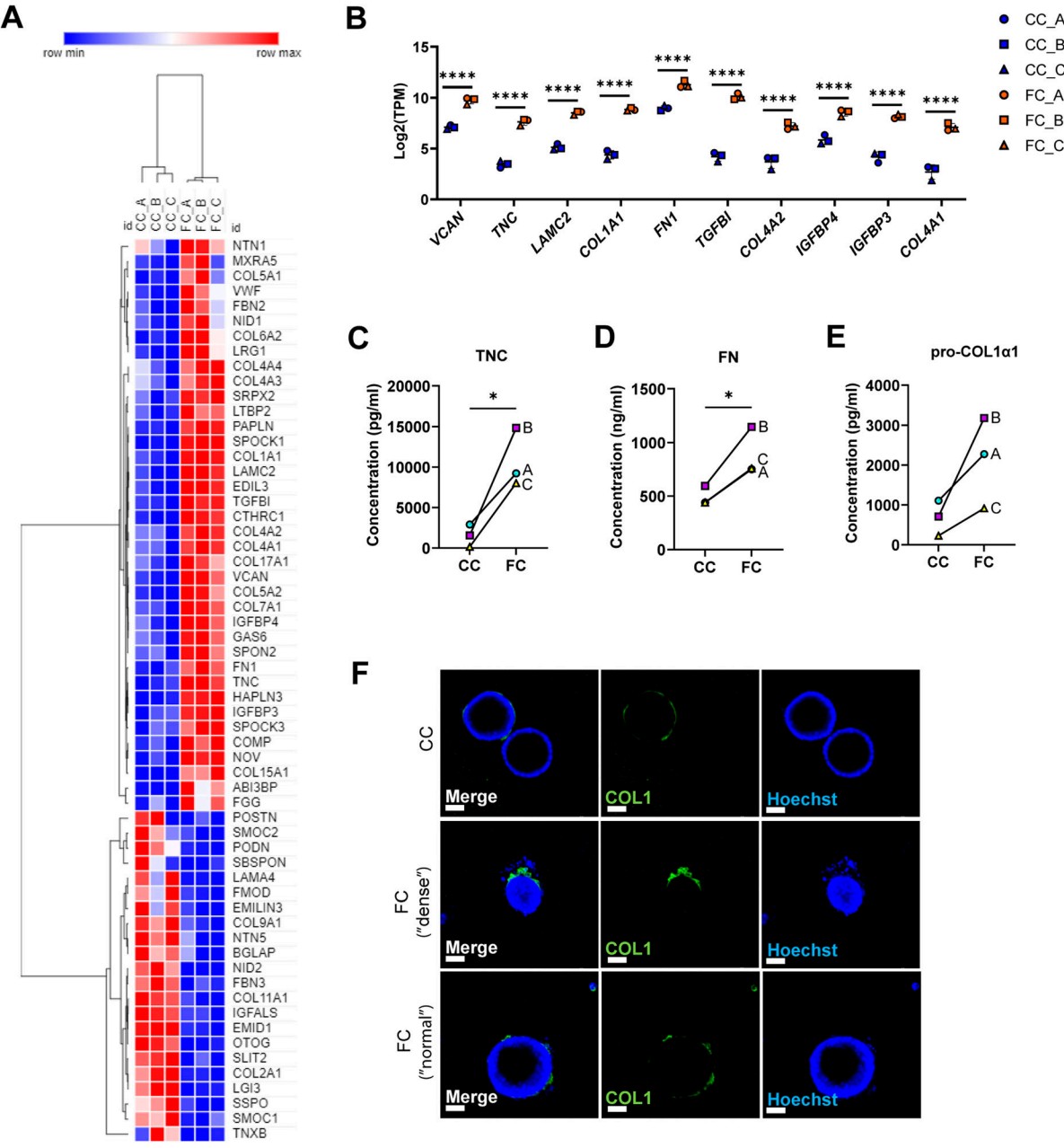

**Figure 6. Fibrosis cocktail induces production of ECM.**
**(A)** Heatmap of significantly dysregulated matrisome genes in FC-stimulated alveolospheres by absolute fold change (*P(adj)* < 0.05). **(B)** Selected matrisome genes from the Core Matrisome gene set in Table 1 in Naba et al (57) with the highest expression in FC-stimulated alveolospheres (*P(adj)* < 0.05, log$_2$[TPM] ≥ 5, log$_2$ fold change ≥ 2). Data presented as means ± SD. **** = *P(adj)* < 0.0001. *n* = 3 batches of alveolospheres. **(C)** Secreted concentrations of tenascin C measured by ELISA. * = *P* < 0.05 by paired, two-tailed *t*-test. **(D)** Secreted concentrations of fibronectin measured by ELISA. * = *P* < 0.05 by paired, two-tailed *t*-test. **(E)** Secreted concentrations of pro-collagen 1α1 measured by ELISA. Paired, two-tailed *t*-test performed. **(F)** Immunofluorescence of alveolospheres stimulated with CC or FC under non-permeabilised conditions. Scale bar = 50 *µ*m. See also Fig S10.

and of the reprogramming-associated markers *KRT8* and *KRT17* after treatment (Fig S15B). However, none of the drugs prevented the loss of surfactant protein expression which occurred after FC treatment (Fig 8I and J), indicating that the AEC2 phenotype was not rescued by the treatment. The acquisition of the mesenchymal and stromal markers induced by the FC were not because of the loss of *NKX2-1* or *CDH1* expression indicating preserved lung epithelial cell

phenotype in the alveolospheres (Fig S16). Altogether, these data show that the alveolospheres respond to anti-fibrotic therapy, neither pirfenidone nor nintedanib can fully restore the alveolospheres to their pre-injury phenotype. Thus, our model holds potential to serve as a relevant model for drug discovery in the future to identify compounds which can fully prevent further damage to the epithelium or even reverse existing damage.

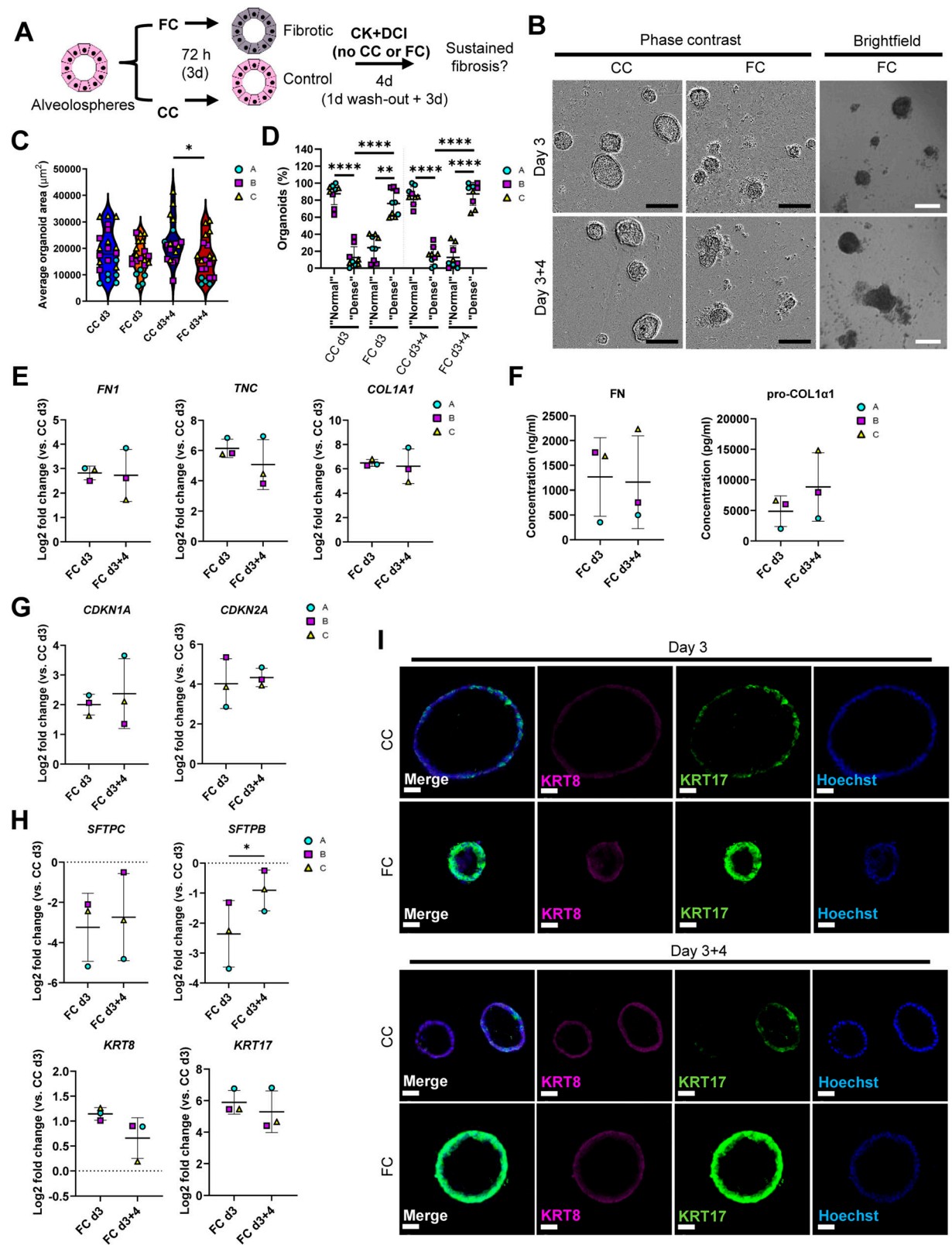

**Figure 7. Effects of the fibrosis cocktail persist after withdrawal.**
**(A)** Schematic showing FC stimulation of alveolospheres to assess persistence of fibrotic responses. **(B)** Representative phase contrast and brightfield images of alveolospheres stimulated with the CC or FC for 3 d and after withdrawal of the CC or FC for additional 4 d. 4x magnification. Scale bar = 250 $\mu$m for phase contrast, 200 $\mu$m for brightfield. **(C)** Average organoid areas of alveolospheres stimulated with either CC or FC and after withdrawal. * = $P < 0.05$ by one-way ANOVA with Sidak's multiple

## Discussion

Our aim was to create a model of the transitional/intermediate alveolar epithelial cell types arising in IPF. We have shown that several aspects of IPF can be modeled in vitro by exposing alveolospheres to a pro-fibrotic milieu mimicked by the FC. A major advantage of iAEC2 is that they can be derived from accessible adult somatic cells (17, 18, 58), which does not require organ donation and is of less ethical concern than the isolation of ESC. Although iAEC2 are considered to have a less mature phenotype than their primary counterparts, mainly considering their expression of immune-related genes (59), they share key phenotypic properties associated with primary AEC2 such as surfactant processing in lamellar bodies. This makes iAEC2 a suitable source for modeling diseases involving this cell type (18). As the iAEC2 in our alveolosphere model have been derived in a microenvironment which is different from the one that primary AEC2 are exposed to in individuals with IPF, our model can potentially give insights into the mechanisms of early fibrosis initiation which is challenging in models utilising primary patient-derived cells from the lung tissue. In addition, one of the main benefits of iAEC2 is their considerably higher proliferative capacity compared with primary AEC2 (18). The expandability of the iAEC2 in the miniaturised culture format is a significant benefit as it enables our system to be further adapted for use in drug screening and precise genome editing using technologies such as CRISPR/Cas9 (60) in the future. We have used the same concentrations of the components of the FC as reported previously (27) which have shown effects on various cell types in PCLS from several human lung tissue donors (28). Donor-associated variability in responses to the FC has however been observed (27). We have focused on evaluating the consistency in response to the FC across several differentiations (batches) of our iPSC line. The fact that the effects of the FC can be reproduced in the different batches and in several experiments as demonstrated in this study is important for the future perspective of adapting our system for applications including drug screening, as such screens are commonly performed using one cell line. However, the evaluation of several iPSC donors has not been undertaken in our study. This is an important future perspective which would need to include donors ranging in age, gender, and ethnicity to fully recapitulate the potential differences in responses to the FC which could yield additional information relevant to IPF.

In our model, we have been able to study the effects of the FC directly on cells resembling human alveolar type 2 cells (i.e., iAEC2) in the absence of other supporting cells, which is a challenging aspect to study in systems like primary adult human distal lung organoids or the ex vivo PCLS model. Several markers described to be differentially regulated in our alveolosphere model were also originally described to be affected by the FC in the human PCLS model (27). This includes up-regulated expression of fibronectin and collagen type 1 with simultaneous down-regulation of surfactant protein C. However, because of the complex multicellular nature of the PCLS system, it was unclear whether the FC acted directly on the epithelial cells or whether the observed changes in alveolar epithelial markers were a result of factors secreted by neighbouring cells. Furthermore, whether or not epithelial cells directly contributed to up-regulation of matrix components in that model was unknown. Therefore, by using an alveolosphere model of iAEC2, we showed that iAEC2 are capable of producing ECM components upon FC stimulation.

Although we could see increased levels of released LDH in the medium which may be indicative of induced cell death by the FC, this could also be because of induced dysfunction in lactate metabolism as has been described for the fibrotic alveolar epithelium and in animal models of pulmonary fibrosis (61, 62). Dysfunctional lactate metabolism may also explain the significant differences in transcript levels of some of the LDH subunits induced with the FC (up-regulated *LDHA*, down-regulated *LDHB* and *LDHD*, as outlined in SdataF2.1). Intriguingly, pathway analysis of our bulk RNA-seq dataset identified a number of activated pathways which have also been described to be activated in iAEC2 derived from interstitial lung disease patients harbouring mutations in the *SFTPC* gene, including apoptosis, the p53 pathway, and PI3K-Akt signaling pathway (63).

Although studies using iPSC derived from patients with IPF or familial pulmonary fibrosis enable the identification of genetic or epigenetic disease drivers (64), our approach of using iPSC not specifically derived from IPF patients is therefore different, as it focuses on factors present in the microenvironment which can contribute to the aberrant fibrotic reprogramming of the iAEC2. Although our study focuses on the extrinsic factors possibly contributing to the initiation of IPF, it is also possible that the FC stimulation models certain responses which are common to familial pulmonary fibrosis as well. Another study has similarly explored the possibility of inducing fibrotic changes with an IPF-relevant cocktail (IPF-RC) as has yet another study which used TGF-β1 stimulation alone in an iPSC-derived alveolar air–liquid interface model (24, 65). However, the IPF-RC differs to the FC in its composition and concentrations and required a considerably longer stimulation of 15 d. Such long stimulations would not be desirable for rapid high throughput drug-screening assays because of the high cost and because of the fact that longer culture time can spontaneously induce changes in cellular phenotype even without injury (53). In addition, the expression of *COL1A1*, which is central to IPF pathology,

---

comparisons test. *n* = 3–8 independent images per batch from three batches of alveolospheres. **(D)** Percentage of organoids with "normal" and "dense" morphologies in brightfield images of CC- and FC-stimulated cultures at 3 d of stimulation and after withdrawal of the CC or FC for an additional 4 d. **** = *P* < 0.0001, ** = *P* < 0.01 by repeated measures one-way ANOVA with Sidak's multiple comparisons test. *n* = 3 independent images per batch and time point from three batches of alveolospheres. **(E)** Expression of genes measured by qRT–PCR related to ECM production. **(F)** Secreted protein concentrations of fibronectin and pro-collagen 1α1 in the medium measured by ELISA. Data presented as means ± SD. Paired, two-tailed *t*-test performed. *n* = 3 batches of alveolospheres. **(G)** Expression of genes measured by qRT–PCR related to cell cycle arrest. **(H)** Expression of genes measured by qRT–PCR related to alveolar epithelial injury and reprogramming. **(I)** Immunofluorescence of alveolospheres stimulated with CC or FC after 3 d of stimulation and after an additional 4 d of CC or FC withdrawal. Scale bar = 50 μm. For (E, G, H): Expression of genes normalised to the average CT value of the reference genes *GAPDH*, *TBP*, and *HPRT1*. Fold changes (2^−ΔΔCT) calculated by comparison with the average ΔCT value of the CC day 3 population. Data presented as means ± SD. Significance tested by paired, two-tailed *t*-test. *n* = 3 batches of alveolospheres. See also Figs S11 and S12.

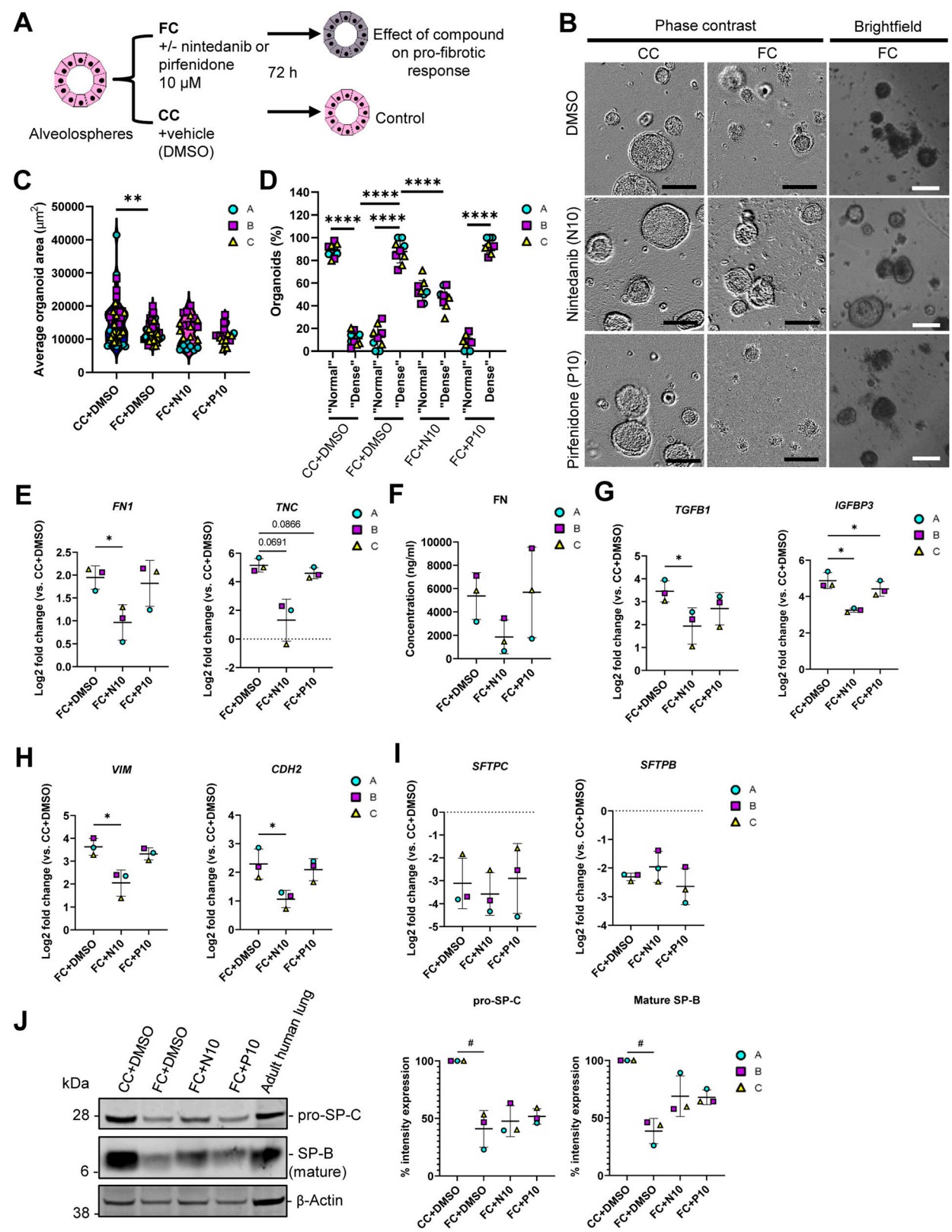

**Figure 8. Fibrotic alveolospheres respond to anti-fibrotic treatment.**
**(A)** Schematic depicting prophylactic treatment of FC-stimulated alveolospheres with nintedanib or pirfenidone. **(B)** Representative phase contrast and brightfield images of alveolospheres stimulated with the CC or FC for 72 h with or without 10 μM nintedanib or pirfenidone. 4x magnification. Scale bar = 250 μm for phase contrast, 200 μm for brightfield. **(C)** Average organoid areas of alveolospheres stimulated with either CC or FC with or without 10 μM nintedanib or pirfenidone. ** = $P < 0.01$ by one-

was not altered after stimulation with the IPF-RC (24), and not investigated with the TGF-β1 stimulation (65). Whether or not the IPF-RC or TGF-β1 alone can induce epithelial reprogramming seen in IPF, such as the appearance of aberrant basaloid cells, remains unknown. Future studies comparing the different modes of injury across the same cell types and models are critical to understanding the contribution of each component inducing the reprogramming we observe in our model and the mechanism behind the FC.

One important aspect of our model is that we utilise 3D alveolospheres which allows the assessment of morphological changes such as organoid size and density. This could be included as a label-free phenotypic readout in future drug screens. The evaluation of such morphological changes is not possible in air–liquid interface models (24, 65). The presence of organoids with the "normal" and "dense" morphologies in the FC-stimulated cultures is possibly a contributing factor to the variation in some of our readouts, as these two morphologies may potentially be associated with different cell phenotypes. We provide some information regarding the visible differences in KRT17 expression in these two types of morphologies in this study. However, more detailed analyses of the transcriptomic differences associated with the "normal" and "dense" morphologies are needed to determine whether there are actual differences in cell phenotypes, which is an interesting future perspective.

The shifts in the distribution and identities of lung epithelial cells in IPF is a relatively recent observation due largely to the availability of single-cell RNA-seq data from human IPF lungs (5, 8, 9, 10). Thus, models in which these aberrant cell identities can be studied are only beginning to emerge. In our study, we have applied computational deconvolution of our bulk RNA-seq dataset compared with public single-cell RNA-seq datasets from IPF patients (10) and in vitro alveolar organoid cultures (53) to obtain information about the changes in cellular composition with the FC. Such analyses give us the possibility to predict the induction of interesting and rare cell types but do not provide a definite confirmation of their presence in the culture. In our study, we have been able to confirm the presence of the predicted $KRT5^-/KRT17^+$ aberrant basaloid-like cells in our cultures. An interesting future perspective would be to also confirm whether other predicted cell types, such as ciliated cells, are present in the alveolospheres using alternative techniques at the protein or morphological level.

Other studies have modeled fibrotic alveolar epithelial injury in ESC/iPSC-derived alveolar organoids (19, 21, 22, 23, 53), but these models required co-cultures with non-epithelial cells which limits the possibility to study the effects of defined factors on alveolar epithelial cells, as is possible with the FC in our model. The additional benefit of our mesenchyme-free organoid system is that it does not require isolation of primary cells from the human lung for the culture, hence being more accessible for use at institutions lacking access to human lung tissue. The mesenchyme-free system also eliminates the variability associated with primary cells or tissue material. Currently, the only approved therapeutics for IPF, which mainly target the lung mesenchyme, are unable to reverse disease progression. An alternative therapeutic approach to targeting the mesenchyme directly is to target the epithelium in IPF. This is an attractive alternative as aberrant epithelium may secrete factors, such as TGF-β, which stimulate the overproduction of ECM by mesenchymal cells in IPF. Furthermore, aberrant epithelium may have altered responses to pathogens or other environmental insults (e.g., particle exposure or pollution) which may lead to exacerbations in IPF (e.g., through secretion of pro-fibrotic mediators such as TGF-β). Thus, restoration of the alveolar epithelium is a potentially attractive alternative approach to explore new therapies. Thus, there is a need for alternative model systems such as ours in which modeling of cell types other than the mesenchymal cells is possible to identify potential novel and more effective therapies for IPF. Our model system as presented in our study will be particularly well suited for evaluation of such compounds.

For the modeling of the newly described KRT17⁺ cells specifically, the previous iAEC2 or primary AEC2-based models required co-culture with mesenchymal cells and considerably longer culture time to induce these phenotypic changes as compared with the FC stimulation in our model (53, 59). The fact that the FC induces KRT17 expression similarly to that seen with mesenchymal cell co-culture indicates that the FC contains factors which, to some extent, are able to mimic parts of the mesenchymal signaling during the aberrant alveolar reprogramming associated with IPF. The appearance of the aberrant basaloid cell state has been described as a consistent feature in IPF which is not observed in normal lung epithelium (66). The ability of the FC to induce this cell state is therefore an encouraging observation. The FC also induces expression of genes of known biomarkers in IPF such as *MMP7* (36) and *IGFBP3* (67), and a pro-fibrotic signature including the secretion

way ANOVA with Dunnett's multiple comparisons test (comparisons to FC+DMSO). $n$ = 8–16 independent images from each of three batches and for each condition of alveolospheres. **(D)** Percentage of organoids with "normal" and "dense" morphologies in brightfield images of alveolospheres stimulated with CC or FC for 72 h with or without 10 μM nintedanib or pirfenidone. **** = $P$ < 0.0001 by repeated measures one-way ANOVA with Sidak's multiple comparisons test. $n$ = 3 independent images per batch and time point from three batches of alveolospheres. **(E)** Expression of genes measured by qRT–PCR related to ECM production. * = $P$ < 0.05. **(F)** Secreted protein concentrations of fibronectin in the medium measured by ELISA. Data presented as means ± SD. Repeated measures one-way ANOVA (comparisons with FC+DMSO) performed. $n$ = 3 batches of alveolospheres. **(G)** Expression of genes measured by qRT–PCR related to pro-fibrotic signaling. * = $P$ < 0.05. **(H)** Expression of genes measured by qRT–PCR related to alveolar epithelial reprogramming. * = $P$ < 0.05. **(I)** Expression of genes measured by qRT–PCR related to alveolar epithelial injury.

**(J)** Representative Western blot of intracellular protein lysates from alveolospheres stimulated with CC+DMSO or FC+DMSO and treated with either nintedanib (FC+N10) or pirfenidone (FC+P10). Adult human lung tissue lysate is positive control. The signal was quantified as band intensity normalised to β-actin and expressed as percentages compared with the respective CC+DMSO sample which is set to 100% expression. Data presented as means ± SD (FC+DMSO, FC+N10 and FC+P10). # = $P$ < 0.05 by one sample $t$-test on the percentage intensity compared with a hypothetical value of 100 (CC+DMSO versus FC+DMSO) and repeated measures one-way ANOVA test were performed (FC+DMSO versus FC+N10 or FC+P10). $n$ = 3 batches of alveolospheres. Blots are cropped from the same membrane and are obtained by sequential blotting as outlined in Supplemental Data 1. For (E, G, H, I): expression of genes normalised to the average CT value of the reference genes *GAPDH*, *TBP*, and *HPRT1*. Fold changes ($2^{-\Delta\Delta CT}$) calculated by comparison with the average ΔCT value of the CC+DMSO population. Data presented as means ± SD. Significance tested by repeated measures one-way ANOVA, if significant with Dunnett's multiple comparisons test (comparisons to FC+DMSO). $n$ = 3 batches of alveolospheres. See also Figs S13–S16.

of ECM and aberrant expression of mesenchymal markers. In particular, the observation that the FC induced an aberrant basaloid-like phenotype in our alveolosphere system will allow us to study the induction of this rare population by using defined stimulants without the need of co-cultures.

Our study indicates that the aberrant epithelial subpopulations present in the IPF lung may arise through rapid reprogramming of the distal epithelium. Thus, our study provides additional insight into the potential origin of the aberrant basaloid cells, which have also been suggested to arise from airway epithelial cells (68, 69, 70) or through migration of proximal epithelial progenitors into the distal airspaces (71). Notably, however, some of these published studies rely on cells isolated from patients with already established IPF (68, 70). These cells may have already undergone aberrant reprogramming or activation in vivo before isolation, which potentially obscures the interpretation of the observed results when these cells are cultured in vitro.

We also validated that our model is responsive to anti-fibrotic treatment by using a prophylactic treatment approach with the two approved therapies nintedanib and pirfenidone. Although prophylaxis in the clinic is a challenging concept as IPF is not detected at onset, this rationale could be appropriate for preventing progression of injury or after exacerbations. Nintedanib significantly prevented the induction of the dense morphology by the FC in the alveolospheres, which was not seen after treatment with pirfenidone. This may possibly be because of insufficient concentrations of pirfenidone in our system, which we were unable to increase further because of risk of cytotoxicity induction. However, other studies have also described more favourable effects of nintedanib treatment in fibrotic alveolar epithelial cells compared with pirfenidone treatment, even when higher concentrations of pirfenidone were used (28). Although further experiments are needed to confirm the molecular mechanism behind the effect on organoid morphology with nintedanib, a possible explanation could be related to the ability of nintedanib to act on processes related to TGF-$\beta$ signaling, a component of the FC (72). This would be consistent with other in vitro studies which have demonstrated that inhibition of TGF-$\beta$ signaling in fibrotic alveolar organoids containing fibroblasts mediates morphological effects visible through reduced contraction and less densening (21). It is however possible that the effect on the dense phenotype is attributed to the fibroblasts present in that system, and these are lacking in our model. This is one example of when the determination of which effects are attributed specifically to the alveolar epithelial cells becomes challenging in multicellular models. Our mesenchyme-free alveolosphere model can provide or help experimentally dissect these aspects, which will be the key to pursue in future studies for further understanding the disease itself, and also to enable specific therapeutic targeting of alveolar epithelial cells in IPF. Interestingly, although we observed effects of nintedanib on markers associated with fibrosis and EMT, in line with published studies (28, 73), we did not observe any rescue of surfactant protein expression. This is in contrast to other studies using primary AEC2 from bleomycin-injured mice and human PCLS (28). The variation in responses could be because of the difference in species, culture on plastic as opposed to Matrigel, and effects of other cell types present in the PCLS. We have in this study given an overview of the responses to

anti-fibrotic therapy as a proof of concept that our system has the potential to be adapted for use in drug screening in the future, and our results highlight that there is a need for novel anti-fibrotic compounds which target the alveolar epithelium (74).

In conclusion, we have shown that stimulation of alveolospheres with the FC is a system that models key features of human IPF. The model is suitable for studying fibrotic alveolar epithelial reprogramming and shows potential for use in drug discovery.

# Materials and Methods

For more detailed descriptions of experimental procedures, please see Supplemental Information.

## Cells and human lung tissue

The r-iPSC1J cell line was reprogrammed from newborn foreskin BJ fibroblasts (CRL2522; ATCC) acquired by AstraZeneca in compliance with the ATCC materials transfer agreement as previously described (75). The ethical consent form for the BJ fibroblast line was requested from ATCC by AstraZeneca but was not available; therefore, the generation of the r-iPSC1J cell line was reviewed and supported by the AstraZeneca Human Biological Sample Governance Team.

The use of human lung tissue was approved by the local ethics committee in Lund, Sweden (Regionala Etikprövningsnämnden, Lund), Dnr 2013/253. All donors provided informed consent and the experiments were conducted in accordance with the Declaration of Helsinki.

## Fibrosis cocktail stimulation of alveolospheres

Thawed alveolospheres in passages 2–3 were resuspended and cultured in growth factor-reduced Matrigel (354230; Corning) in 96-well plates with CK+DCI medium (Tables S1 and S4) as described previously (17, 18) for 2–7 d before the first single-cell passage. Single iAEC2 in passages 4–6 (two to three passages post thaw) were plated at a density of 100 cells/$\mu$l and formed organoids for 14 d. To induce fibrosis, alveolospheres were stimulated with the addition of the FC consisting of 5 ng/ml recombinant TGF-$\beta$ (240-B-002/CF; R&D Systems), 10 ng/ml PDGF-AB (PHG0134; GIBCO), 10 ng/ml TNF-$\alpha$ (410-MT-010/CF; GIBCO), and 5 $\mu$M lysophosphatidic acid (62215; Cayman Chemical) or the CC (diluents of components) to the CK+DCI medium for 72 h. For the FC withdrawal experiment, iAEC2 were plated at a density of 50 cells/$\mu$l to avoid overgrowth because of longer culture time. For drug treatment, alveolospheres were stimulated with the FC with simultaneous addition of DMSO (control) and either nintedanib (FM34144; Carbosynth) or pirfenidone (FM341441402; Carbosynth) in concentrations 0.1–10 $\mu$M for 72 h.

## Statistics

The *n* numbers represent samples from differentiations performed at separate time points (batches named A, B, and C) unless

otherwise stated and data are presented as mean ± SD or as described in figure legends. The statistical tests are described separately in the figure legends. *P*-values of < 0.05 were considered significant. Statistical tests and graphs were generated using GraphPad Prism v8 unless otherwise stated.

## Data Availability

The RNA-seq data generated in this study are deposited in ArrayExpress with accession ID: E-MTAB-11676. All code used for the re-processing and deconvolution of the dataset with GEO Accession: GSE150068 is deposited in the GitHub repository at: https://github.com/Lung-bioengineering-regeneration-lab/FC_alveolospheres.

## Supplementary Information

## Acknowledgements

We are grateful to Johan Mattsson and Ulf Gehrmann (AstraZeneca) for performing the human lung tissue lysis. We are thankful to Anette Persson-Kry and Louise Stjernborg (AstraZeneca) for providing and maintaining the iPSC. We thank Source Bioscience for the library preparation and sequencing services. Finally, we thank the donors of the lung tissue and the Sahlgrenska University Hospital for technical support. The Knut and Alice Wallenberg foundation, the Medical Faculty at Lund University, and Region Skåne are acknowledged for generous financial support (DE Wagner). This project is partially funded by the Swedish Research Council Starting Grant (DE Wagner Dnr 2018-02352).

### Author Contributions

V Ptasinski: conceptualization, formal analysis, validation, investigation, visualization, methodology, and writing—original draft, review, and editing.
SJ Monkley: formal analysis, methodology, and writing—review and editing.
K Öst: formal analysis, validation, investigation, visualization, methodology, and writing—review and editing.
M Tammia: formal analysis, investigation, visualization, methodology, and writing—review and editing.
HN Alsafadi: formal analysis, visualization, methodology, and writing—review and editing.
C Overed-Sayer: investigation, methodology, and writing—review and editing.
P Hazon: conceptualization, supervision, project administration, and writing—original draft, review, and editing.
DE Wagner: conceptualization, supervision, funding acquisition, project administration, and writing—original draft, review, and editing.
LA Murray: conceptualization, supervision, funding acquisition, project administration, and writing—original draft, review, and editing.

### Conflict of Interest Statement

AstraZeneca funded this study and participated in the study design, data collection, data analysis and data interpretation. V Ptasinski, SJ Monkley, K Öst, M Tammia, C Overed-Sayer, P Hazon, and LA Murray were full-time employees at AstraZeneca at the time of the study and may own shares in AstraZeneca.

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
