## [Reviewer comments · Life Science Alliance]

Life Science Alliance

Modeling Fibrotic Alveolar Transitional Cells with Pluripotent Stem Cell-derived Alveolar Organoids

Lynne Murray, Victoria Ptasinski, Susan Monkley, Karolina Ost, Markus Tammia, Hani Alsafadi, Catherine Overed-Sayer, Petra Hazon, and Darcy Wagner

DOI: <https://doi.org/10.26508/lsa.202201853>

Corresponding author(s): Lynne Murray, MiroBio and Darcy Wagner, Lund U

Review Timeline:	Submission Date:	2022-11-25
	Editorial Decision:	2022-12-19
	Revision Received:	2023-03-17
	Editorial Decision:	2023-04-05
	Revision Received:	2023-05-09
	Accepted:	2023-05-10

Scientific Editor: Novella Guidi

Transaction Report:

December 19, 2022

Re: Life Science Alliance manuscript #LSA-2022-01853-T

Dr. Lynne Anne Murray
MiroBio
Winchester House
Oxford Science Park
Oxford, Oxfordshire OX4 4GE
UNITED KINGDOM

Dear Dr. Murray,

Thank you for submitting your manuscript entitled "Modelling Aberrant Epithelial Reprogramming in IPF using iPS-derived Alveolar Organoids" to Life Science Alliance. The manuscript was assessed by expert reviewers, whose comments are appended to this letter. We invite you to submit a revised manuscript addressing the Reviewer comments.

Thank you for this interesting contribution to Life Science Alliance. We are looking forward to receiving your revised manuscript.

Sincerely,

B. MANUSCRIPT ORGANIZATION AND FORMATTING:

Reviewer #1 (Comments to the Authors (Required)):

Ptasinski et al., have made use of alveolospheres derived from iPSCs by differentiating these cells into AEC2s. They quite convincingly show the expression of lung progenitor marker NKX2.1. The differentiation into alveolospheres is shown by expression of SP-C and SP-B (imaging, WB analysis). And the preservation of Pro-SP-C expression after cryopreservation is also well documented and convincing. Authors have performed bulk RNA seq analysis of the said alveolospheres and deconvoluted this data into single cell data using existing single cell data publicly available. This comparison predicts that indeed these organoids, express distal lung epithelium genes as well as different cell types. The authors propose to use alveolospheres as a screening model system for factors (drugs, biomolecules etc) that would ameliorate epithelial cell senescence, epithelial cell injury in a prophylactic manner in vivo. And rightly so, the authors have been successful in creating these phenotypes (at the mRNA level, RNA seq and qPCR analysis, to some extent with microscopy) using treatment of these organoids with a mixture of pro-inflammatory, pro-fibrotic factors (FC- TGF β , PDGF-AB, TNF- α and LPA). The pathway analysis as well as comparison to IPF derived data sets, nicely show regulation of fibrotic process upon treatment with FC. Impressively authors show the presence of pro-fibrotic transient cell states KRT5-/KRT17+, and upregulation of major fibrotic markers such as those of MMP7, SERPINE1, IGFBP3 and others. The use of Nintedanib and pirfenidone to ameliorate pro-fibrotic phenotypes and basis for use in screening platform is well justified.

In general, I think the data presentation, data interpretation is well done. The authors have adequately cited existing work by other colleagues in this direction. However, I also believe that the manuscript may improve if the authors could perform following experiments to further the claims in the paper.

Major comments:

1) The authors show that the organoids undergo a decrease in size upon treatment with FC (compared to CC treated organoids). It is not clear if this is due to cell death, or because of induction of cellular senescent pathways. To clarify this point, it might be helpful if authors could make LDH measurements from media supernatants of CC and FC treated organoids over time. Measure LDH prior to FC addition (1 time point) and after (couple of time points- early and 72 hrs post FC treatment). What could also help is to measure number of cells (using simple Hoechst staining) before and after FC treatment, doesn't need to be live imaging but indirect measurements after chemical fixation would also be helpful. This data, in my opinion would help to clarify if the drop in size is due to epithelial cell injury (cell death by FC) or indeed by eliciting cell senescence, although this point (cell senescence) has been shown by expression of senescence markers but the cell death issue needs to be addressed.

2) Authors write that after FC treatment they seem to have 2 populations of alveolospheres, with dense and "normal-looking" morphology (line 238 and other places as well). Given authors propose to use this model system for screening, I think it is imperative to know the variability in the existence of these phenotypes. I think it might help to measure and show the percentage of the dense and normal looking organoids after FC treatment.

I see that the data (qPCR at least) derived from organoids with FC treatment is quite variable in general, and one of the confounding factors may be that dense and "normal-looking" morphology organoids are cell state wise or cellular signaling wise different.

As these could indeed be two states of alveolar phenotypes derived by the same treatment of FC, and this difference of phenotypes may yield additional information if followed using molecular techniques using RNA seq etc.

The comments in bold may simply be discussed in the manuscript.

3) Authors claim deposition of collagen in the ECM of organoids after FC treatment (Figure 5F), unfortunately, the resolution at which the images have been taken and the fact that these are detergent permeabilized (assuming, as the materials and methods section explanation is missing) organoids, this is not the optimal experiment set-up to measure ECM deposition of COL1. I suggest if authors want to claim deposition of COL1 in ECM, COL1 staining in non-permeabilized organoids (plus minus FC) needs to be done. Although, the secretion of more collagen, TNC and FN is documented in the form of ELISA (Figure 5C-E).

- 4) Please list in the materials and methods following information
- List of antibodies used for imaging and western blots with the used dilutions
 - qPCR primer sequences (or references of papers, in case already published)
 - ELISA kits used and measurement protocols
 - Microscopes used for imaging, methods (image analysis platform) used for measuring area of organoids

Minor comments:

- The authors have used a "senescence-associated secretory phenotype (SASP) factors under FC stimulation (Fig. S3B)" gene list, a closer look into, it seems the list is ECM/ Matrisome gene list but no a SASP list. Kindly check into this.
- Line 244, the claim not to have observed any increase in other AEC1 characteristics markers such as AGER, while true (for PDPN), but for AGER the data suggest that it is significantly decreased (Figure S5B). Please write it as such.
- Authors have used deconvolution method to predict presence of many cells types in the organoid culture conditions, it may be beyond the scope of this manuscript but to mention in future to check the existence of these cells (by FACS and other methods) might help. For instance it would great to see if there are actual ciliated cells present in these alveolospheres. This could though be discussed briefly in the manuscript.

Reviewer #2 (Comments to the Authors (Required)):

Ptasinski et al. developed an in vitro model to study IPF by applying a fibrosis cocktail (FC) to treat IPS-derived alveolar organoids. Further, the authors did a bulk RNAseq comparing FC treated organoids with control cocktail (CC) treated organoids. The authors further deconvoluted the bulk RNA-seq data using publicly available scRNA-seq data as references and claim to observe IPF-like aberrant epithelial changes in this model. Overall, the model and the data are very promising, analysis is very thorough. However, due to the nature of bulk RNAseq and the methods of deconvolution to interpret the data for cell type changes, the conclusion drawn from these analyses needs to be cautious.

Some other concerns:

- KRT17 is expressed by all Basal cells, not limited to so-called basaloid cells. Thus, the immunostaining for organoids when the authors tried to show the aberrant basaloid, the authors should include more markers, as they are negative for KRT5.
- Similar for KRT8 staining for transient AEC2-AEC1 cells. Using KRT8 IF to identify these cells actually is problematic, because these cells are identified at the transcript level. But KRT8 a lot of time the translation level is disconnected from its transcription level, it can be detected in many epithelial cells (including both airway and alveolar cells) by IF without transcript levels. Thus, the more accurate of these cells should be described as KRT8 high rather than KRT8+.
- The model in this manuscript is to apply a fibrosis cocktail to normal IPS derived alveolar organoids to induce an IPF-like signature. It is very interesting and provides a new in vitro tool for the community. The authors may possibly make more comments on their model comparing with iACE2-derived alveolar organoids from IPF patient (IPS cells from IPF patient) from Alysandratos et al. (PMID: 36454643).
- Another minor concern is using SPB as AEC2 cell marker. SPB is also expressed by non AEC2 cells like the SCGB3A2+ secretory cells.

Reviewer #3 (Comments to the Authors (Required)):

The manuscript by Ptasinski and colleagues addresses a very relevant issue in aiming to contribute with their work to expansion of in vitro fibrosis models. The researchers investigate the option of using human-derived induced pluripotent stem cells and stimulate the differentiated alveolar type-2 cells with a published fibrosis cocktail. They perform an extensive characterization of the cells in culture and of the resulting response upon exposure to the fibrosis cocktail. Finally they include anti-fibrotic drugs to evaluate the potential for drug screening.

Overall their results support the establishment of an epithelial only fibrosis model, a perhaps relatively small but valuable step forward in development of new representative models. The authors however also propose their model to be relevant for drug screening and this is in my view not sufficiently convincing.

I have several comments or questions related to this conclusion:

- A larger limitation of the study is that the model includes only alveolar type-2 epithelial cells, which is not representative of the alveolar (epithelial) compartment, or the only cell-type that is key in the fibrosis process. The authors mention this platform to be

relevant for drug screening, however the current model poses of course the limitation that many compounds that will be tested aim to reduce the contribution of the mesenchymal compartment to the fibrosis process, which cannot be tested in this system. Generally the aim of an anti-fibrotic drug is to limit the processes that in the body produce the fibrosis cocktail and not per se to alter the response of the epithelium to this cocktail. It is therefore not clear how the authors envision this model to be a representative anti-fibrotic drug screening tool.

2. The results are all obtained with one cell line and effects can differ markedly between cell lines. Can the authors confirm using a second hiPSC line that main effects of FC can be reproduced?

3. The choice of testing Nintedanib and Pirfenidone in this alveolar type-2 only model is a bit puzzling to me, especially with regard to pirfenidone. Both Nintedanib and Pirfenidone target predominantly the fibroblasts compartment, with Nintedanib additionally affecting VEGFR. Nintedanib could have therefore impacted other cell-types such as endothelial cells, for example impacting angiogenesis, but their effects on epithelial (type-2) cells should be limited.

4. Can the authors provide some clarity about the purity of the organoids? Can the authors for example stain the organoids with KRT5? I am a bit surprised to see a baseline gene expression level of KRT5 in Fig. 3C and wonder what the purity is of the total organoid population in culture. Are all organoids expressing AT2 markers or is a percentage of organoids KRT5-positive as they perhaps went into the direction of airway? KRT17 is also expressed by basal cells and the staining of the CC cultures in Fig. 3D also suggest that within an organoid there may be some basal cells present?

5. Figure 5 indicates polarized collagen I expression according to the authors, but the organoid that is depicted in the image has no lumen, a by the authors called: 'dense' organoid. Can there still be secretion from any other location than basal in that case? So is this indeed polarization or a normal consequence of a missing lumen?

6. Page 7 line 140: was there also a reduction in the number of organoids? Was this assessed? And how was the incubation of 72 h chosen?

7. The authors report no visible changes after FC withdrawal (p15, line 355-357), but the images in Fig. 6G suggest a much larger lumen or is this not a good representation in that respect?

Reviewer 1:

Ptasinski et al., have made use of alveolospheres derived from iPSCs by differentiating these cells into AEC2s. They quite convincingly show the expression of lung progenitor marker NKX2.1. The differentiation into alveolospheres is shown by expression of SP-C and SP-B (imaging, WB analysis). And the preservation of Pro-SP-C expression after cryopreservation is also well documented and convincing. Authors have performed bulk RNA seq analysis of the said alveolospheres and deconvoluted this data into single cell data using existing single cell data publicly available. This comparison predicts that indeed these organoids, express distal lung epithelium genes as well as different cell types. The authors propose to use alveolospheres as a screening model system for factors (drugs, biomolecules etc) that would ameliorate epithelial cell senescence, epithelial cell injury in a prophylactic manner in vivo. And rightly so, the authors have been successful in creating these phenotypes (at the mRNA level, RNA seq and qPCR analysis, to some extent with microscopy) using treatment of these organoids with a mixture of pro-inflammatory, pro-fibrotic factors (FC- TGF β , PDGF-AB, TNF- α and LPA). The pathway analysis as well as comparison to IPF derived data sets, nicely show regulation of fibrotic process upon treatment with FC. Impressively authors show the presence of pro-fibrotic transient cell states KRT5-/KRT17+, and upregulation of major fibrotic markers such as those of MMP7, SERPINE1, IGFBP3 and others. The use of Nintedanib and pirfenidone to ameliorate pro-fibrotic phenotypes and basis for use in screening platform is well justified.

In general, I think the data presentation, data interpretation is well done. The authors have adequately cited existing work by other colleagues in this direction. However, I also believe that the manuscript may improve if the authors could perform following experiments to further the claims in the paper.

Major comments:

1a) The authors show that the organoids undergo a decrease in size upon treatment with FC (compared to CC treated organoids). It is not clear if this is due to cell death, or because of induction of cellular senescent pathways. To clarify this point, it might be helpful if authors could make LDH measurements from media supernatants of CC and FC treated organoids over time. Measure LDH prior to FC addition (1 time point) and after (couple of time points- early and 72 hrs post FC treatment).

R1a: We thank the reviewer for this comment. We have added the requested analyses in Supplemental Figure 4. The LDH measurement confirms that there is increased amounts of LDH in the supernatant over time from the alveolospheres stimulated with the FC which may indicate cell death (Supplemental Figure 4B). This could however also possibly be due to increases in lactate dehydrogenase production or activity itself as has been observed in animal models of pulmonary fibrosis (Judge et al. 2018) and in IPF AEC2s themselves (Newton et al 2021). We have also now re-evaluated our list of genes which are significantly up or down with FC stimulation (Supplemental List S1) and noted that in line with what has been reported in Newton et al. 2021, we also observed upregulation of *LDHA*. This aspect has been added to the discussion of the revised manuscript.

Judge et al. 2018: <https://journals.plos.org/plosone/article?id=10.1371/journal.pone.0197936>

Newton et al. 2021: <https://respiratory-research.biomedcentral.com/articles/10.1186/s12931-021-01866-x>

1b) What could also help is to measure number of cells (using simple Hoechst staining) before and after FC treatment, does not need to be live imaging but indirect measurements after chemical fixation would also be helpful. This data, in my opinion would help to clarify if the drop in size is due to epithelial cells injury (cell death by FC) or indeed by eliciting cell senescence, although this point (cell senescence) has been shown by expression of senescence markers but the cell death issue needs to be addressed.

R1b: As a significant portion of the organoids become dense upon FC stimulation, the individual cells become indistinguishable in the most collapsed organoids leading to technical challenges in obtaining accurate cell nuclei counts. To nonetheless provide some sort of correlation to total cell number, we have instead included measurements of total Hoechst-stained area in the cultures and additionally, we have added the average counts of organoids following the two stimulations.

We did not observe a significant loss of Hoechst staining (Supplemental Figure 4C) or significantly fewer organoids (Supplemental Figure 4D) in the FC-stimulated cultures indicating that even though cell death may be induced by the FC, the morphological effects on the organoids (i.e. smaller) are not simply due to a loss of cells after FC stimulation. This lies well in line with our pathway analyses of our bulk RNA-seq dataset (Figure 2E), showing simultaneous enrichment of genes involved in apoptosis as well as the p53-signaling pathway, which is linked to cellular senescence. We hope that the additional analyses have provided further information as to the diversity of effects elicited by the FC treatment in the alveolospheres.

2) Authors write that after FC treatment they seem to have 2 populations of alveolospheres, with dense and "normal-looking" morphology (line 238 and other places as well). Given authors propose to use this model system for screening, I think it is imperative to know the variability in the existence of these phenotypes. I think it might help to measure and show the percentage of the dense and normal looking organoids after FC treatment.

I see that the data (qPCR at least) derived from organoids with FC treatment is quite variable in general, and one of the confounding factors may be that dense and "normal-looking" morphology organoid are cell state wise or cellular signaling wise different.

As these could indeed be two states of alveolar phenotypes derived by the same treatment of FC, and this difference of phenotypes may yield additional information if followed using molecular techniques using RNA seq etc.

R2: This is an excellent suggestion from the reviewer. We have included quantifications of organoid proportions with the "normal" and "dense" morphology in Fig. 1D in the revised manuscript, along with brightfield microscopy images of the organoid cultures to better visualise these two morphologies. To further build in the aspect of the morphology changes into the revised manuscript, we have added similar quantifications in the experiments presented in the revised Figures 7 and 8. Through these quantifications, we could determine that the organoids retain the "dense" morphology following the withdrawal of the FC (Figure 7D), and we could confirm that the FC-associated induction of the "dense" morphology was prevented by nintedanib treatment (Figure 8D). We thank the reviewer for this suggestion as we feel this has provided more detailed information on the effects of the FC on the morphology, which is a readout which can be built in for future applications such as drug screens.

We also appreciate the suggestion made by the reviewer that the presence of organoids with “normal” and “dense” morphology within the FC-stimulated culture could be one of the factors impacting the variability in some of our readouts. To support the discussion that this could potentially be the case, we have included additional images of FC-stimulated organoids with the “normal” and “dense” morphologies in the immunofluorescence panels (Figure 3C, 4D, 6F and Supplemental Figure 10A) to visually add further information on the variation in expression across these two morphologies. We have also included additional text around the morphology-associated variation in the Discussion and discussed the points suggested by the reviewer.

3) Authors claim deposition of collagen in the ECM of organoids after FC treatment (Figure 5F), unfortunately, the resolution at which the images have been taken and the fact that these are detergent permeabilized (assuming, as the materials and methods section explanation is missing) organoids, this is not the optimal experiment set-up to measure ECM deposition of COL1. I suggest if authors want to claim deposition of COL1 in ECM, COL1 staining in non-permeabilized organoids (plus minus FC) needs to be done. Although, the secretion of more collagen, TNC and FN is documented in the form of ELISA (Figure 5C-E).

R3: We thank the reviewer for pointing this out and agree that performing staining without permeabilisation is necessary to make a more definitive statement about the location of the collagen staining. We have now performed COL1 staining in non-permeabilised organoids and exchanged the immunofluorescence images in the revised Figure 6F. The images of the permeabilised cultures have been moved to Supplemental Figure 10.

We also apologise for the missing methods section in our initial submission. See R4 for further information.

4) Please list in the materials and methods following information

- a. List of antibodies used for imaging and western blots with the used dilutions
- b. qPCR primer sequences (or references of papers, in case already published)
- c. ELISA kits used and measurement protocols
- d. Microscopes used for imaging, methods (image analysis platform) used for measuring area of organoids

R4: Thank you for noting this. We now see that, unfortunately, somehow the submission to Life Science Alliance did not include the Supplemental Experimental procedures section but this was included in our bioRxiv deposit. This is now included for review and we have in the revised version added a sentence referring to this section in the main text to help guide the reader to this information. We apologise for this oversight.

- a. Please see Table S5.
- b. Please see Table S6 (TaqMan qPCR assays are provided as pre-designed primers labelled with probes by the supplier. The Assay ID provided in Table S6 provides the reference to all information, including the binding sites in the gene sequences which are available on the supplier’s website).

- c. Please see section Enzyme-linked immunosorbent assay (ELISA), referring to ready-to-use kits by the supplier (kits include instructions for measurement).
- d. As we have expanded the morphological assessment of the organoids in the revised manuscript, we have included a separate section called Organoid morphology measurements describing this information. Please also see the section Immunofluorescence microscopy.

Minor comments:

1) The authors have used a "senescence-associated secretory phenotype (SASP) factors under FC stimulation (Fig. S3B)" gene list, a closer look into, it seems the list is ECM/ Matrisome gene list but no a SASP list. Kindly check into this.

RM1: We thank the reviewer for this comment. The list shown in the revised Supplemental Figure 6C is based on the SASP list described by Coppé et al. 2010 in their Table 1. This list includes factors that have been described to be significantly altered between non-senescent and senescent cells, and include a range of markers including some ECM or matrisome genes as the reviewer correctly pointed out, but also well recognised SASP mediators including *SERPINE1*, *CDKN1A* and *CDKN2A*. In the revised version, we have moved the citation of the study by Coppé et al. 2010 to directly after the reference to our Supplemental Figure 6C and additionally added it directly into the Supplemental Figure 6C to make it clearer that our heatmap is based on this published work.

Coppé et al 2010: <https://www.ncbi.nlm.nih.gov/pmc/articles/PMC4166495/>

2) Line 244, the claim not to have observed any increase in other AEC1 characteristics markers such as AGER, while true (for PDPN), but for AGER the data suggest that it is significantly decreased (Figure S5B). Please write it as such.

RM2: This sentence has been re-worded as suggested by the reviewer.

3) Authors have used deconvolution method to predict presence of many cells types in the organoid culture conditions, it may be beyond the scope of this manuscript but to mention in future to check the existence of these cells (by FACS and other methods) might help. For instance it would great to see if there are actual ciliated cells present in these alveolospheres. This could though be discussed briefly in the manuscript.

RM3: We fully agree with the reviewer that additional experiments at single cell resolution are needed in the future to more comprehensively understand the effects of the FC. This point has been added in the Discussion section in the revised manuscript.

Reviewer 2:

Ptasinski et al. developed an in vitro model to study IPF by applying a fibrosis cocktail (FC) to treat IPS-derived alveolar organoids. Further, the authors did a bulk RNAseq comparing FC treated organoids with control cocktail (CC) treated organoids. The authors further deconvoluted the bulk RNA-seq data using publicly available scRNA-seq data as references and claim to observe IPF-like aberrant epithelial changes in this model. Overall, the model and the data are very promising, analysis is very thorough. However, due to the nature of bulk RNAseq and the methods of deconvolution to interpret the data for cell type changes, the conclusion drawn from these analyses needs to be cautious.

Some other concerns:

1). KRT17 is expressed by all Basal cells, not limited to so-called basaloid cells. Thus, the immunostaining for organoids when the authors tried to show the aberrant basaloid, the authors should include more markers, as they are negative for KRT5.

R1: We thank the reviewer for this suggestion. We have included additional co-staining of KRT17 and KRT5 in Figure 4D (and Supplemental Figure S3B as positive control for the staining), showing that the KRT17 positive cells do not stain positive for KRT5. In addition, other markers which we have immunostained for previously are present in aberrant basaloids but rarely or lowly expressed in normal basal cells (e.g. VIM and COL1). In both of these instances, the FC notably increases the expression of these markers. Future work should focus on understanding these changes on the single cell level (both transcript and protein level). As was also suggested by Reviewer 1, we have added this to the discussion.

2). Similar for KRT8 staining for transient AEC2-AEC1 cells. Using KRT8 IF to identify these cells actually is problematic, because these cells are identified at the transcript level. But KRT8 a lot of time the translation level is disconnected from its transcription level, it can be detected in many epithelial cells (including both airway and alveolar cells) by IF without transcript levels. Thus, the more accurate of these cells should be described as KRT8 high rather than KRT8+.

R2: Thank you for pointing this out. We agree with the reviewer that it is better to use KRT8 high, as more precisely used in recent publications. We have changed the sentence in the Results section in which we linked KRT8+ cells with the transient AEC1-AEC2 cells and have phrased this as stated below:

“We also observed FC-induced expression of *KRT8*, a marker which has been reported to be highly expressed in a transient AEC2-AEC1 population in IPF”

We have also changed KRT8+ to KRT8 high throughout in the text as suggested by the reviewer.

3). The model in this manuscript is to apply a fibrosis cocktail to normal IPS derived alveolar organoids to induce an IPF-like signature. It is very interesting and provides a new in vitro tool for the community. The authors may possibly make more comments on their model comparing with iACE2-derived alveolar organoids from IPF patient (IPScells from IPF patient) from Alysandratos et al. (PMID: 36454643).

R3: Thank you for the suggestion to incorporate these other studies to broaden our discussion. We have included the mentioned paper in the Discussion section as they also observe the emergence of the transitional cell type seen in end-stage IPF samples via single cell RNA sequencing. We have also

discussed our work in the context of another study, also by Alysandratos et al. (see provided references below):

Alysandratos et al. 2023 <https://pubmed.ncbi.nlm.nih.gov/36454643/>

Alysandratos et al. 2021 <https://www.ncbi.nlm.nih.gov/pmc/articles/PMC8432578/>

Although our cells share some similarities in spontaneous responses with cells derived from patients with interstitial lung disease (ILD) (Alysandratos et al. 2021 PMID: 34469722; which uses iPSCs derived from a patient with a surfactant processing mutation, SFTPC I73T), the experimental approach used in our study is different. Studies using iPSCs derived from patients with IPF, familial pulmonary fibrosis or ILD enable the identification of genetic or epigenetic disease drivers. Our approach of using iPSCs not specifically derived from IPF or ILD patients focuses more on factors present in the microenvironment which can contribute to the aberrant fibrotic reprogramming of the iAEC2s. Therefore, our study provides insights into the possible mechanisms of the initiation of epithelial injury in pulmonary fibrosis. As not all patients have known genetic alterations, understanding the sensitivity of non-diseased AEC2s (or iAEC2s derived from patients with or without IPF/ILD) to a pro-fibrotic environment is a valuable tool for the majority of patients with IPF with no known genetic or somatic mutations.

4) Another minor concern is using SPB as AEC2 cell marker. SPB is also expressed by non AEC2 cells like the SCGB3A2+ secretory cells.

R4: We agree that this can be phrased with better precision. In the revised version, we have removed the wording “feature specific to AEC2s” in the Results section and linked the SP-B expression with other AEC2-characteristic readouts such as expression of pro-SP-C in a cytosolic, vesicular pattern. During the differentiation, our validation of the AEC2-like phenotype in our organoids was done by assessing the expression of several surfactants associated with AEC2s (SP-B and pro-SP-C) both by immunofluorescence (Supplemental Figure 1E and in the revised version, Supplemental Figure 2) and Western blotting (Supplemental Figure 1F). This is in line with the validation criteria recommended for evaluation of AEC2-like phenotypes in stem-cell derived cultures and with the validation published in the original differentiation protocol which we have followed.

Perspective from Michael Beers and Yuben Moodley on assessing the AEC2 phenotype:

<https://pubmed.ncbi.nlm.nih.gov/28326803/>

Originally described differentiation protocol:

Jacob et al. 2017: <https://pubmed.ncbi.nlm.nih.gov/28965766/>

Jacob et al. 2019: <https://pubmed.ncbi.nlm.nih.gov/31732721/>

To additionally validate that these cells are negative for proximal lung epithelial markers, which can also be derived through this protocol using different media compositions (McCauley et al. 2017), we have included another co-staining of the organoids with pro-SP-C and keratin 5 (KRT5) which is associated with basal cells (progenitors for airway epithelial cells) in Supplemental Figure S3. No keratin 5 positive cells were detected in our organoids.

McCauley et al. 2017: <https://pubmed.ncbi.nlm.nih.gov/28366587/>

Reviewer 3:

The manuscript by Ptasinski and colleagues addresses a very relevant issue in aiming to contribute with their work to expansion of in vitro fibrosis models. The researchers investigate the option of using human-derived induced pluripotent stem cells and stimulate the differentiated alveolar type-2 cells with a published fibrosis cocktail. They perform an extensive characterization of the cells in culture and of the resulting response upon exposure to the fibrosis cocktail. Finally they include anti-fibrotic drugs to evaluate the potential for drug screening.

Overall their results support the establishment of an epithelial only fibrosis model, a perhaps relatively small but valuable step forward in development of new representative models. The authors however also propose their model to be relevant for drug screening and this is in my view not sufficiently convincing.

1. A larger limitation of the study is that the model includes only alveolar type-2 epithelial cells, which is not representative of the alveolar (epithelial) compartment, or the only cell-type that is key in the fibrosis process. The authors mention this platform to be relevant for drug screening, however the current model poses of course the limitation that many compounds that will be tested aim to reduce the contribution of the mesenchymal compartment to the fibrosis process, which cannot be tested in this system. Generally the aim of an anti-fibrotic drug is to limit the processes that in the body produce the fibrosis cocktail and not per se to alter the response of the epithelium to this cocktail. It is therefore not clear how the authors envision this model to be a representative anti-fibrotic drug screening tool.

R1: Thank you for highlighting this emerging concept and requesting further clarification. We agree with the reviewer that most of the current approaches converge on targeting the mesenchymal compartment. The 'ideal' target cell type in IPF has changed over the past few decades with failed clinical trials targeting inflammation as well as endothelial cells. We agree it is important to improve the discussion around the rationale for targeting the epithelium as this area has only recently gained traction. Throughout the manuscript, we have modified our statements regarding the application of the model in drug screens and formulated this aspect as a future approach. We hope that this will better balance the aspect of disease modeling for the epithelial injury and reprogramming observed in IPF, which was our main focus in this manuscript. We feel that this model can be further adapted and optimised for use in future drug screens.

We agree with the reviewer that the lack of other cell types such as mesenchymal cells could be a limitation under circumstances where modeling of interactions between different cell types is desired within the primary drug screen. We have discussed this in the Discussion in the revised manuscript.

However, and as the reviewer points out ('limit the processes that in the body produce the fibrosis cocktail'), it is currently not fully established which cell types are most crucial for secreting the components of the pro-fibrotic microenvironment. Some of the published single cell studies indicate that the epithelial cells in IPF have increased expression of some of the FC factors (e.g. *TGFB1*, *PDGFA*). However, immune cells are also observed to be altered in IPF (e.g. macrophages and B cells) and have altered levels of these factors. Therefore, it is presently not known which cell type is the major contributor to the production of these pro-fibrotic and pro-inflammatory cytokines.

Interestingly, our data indicates a potential feed forward loop with increased transcriptional levels of *TGFB1*, *TNF*, and *PDGFB* and *PDGFA*. Therefore, targeting the regulation of these factors in alveolar

epithelial cell types is a potentially novel approach in order to restrict further epithelial damage and disease progression in the alveolar environment.

The currently approved therapies mainly aim to target the behaviour of mesenchymal cells and thus far have been shown to be ineffective at reversing disease course or substantially lengthening patient survival more than a few years. There is thus a clear need for alternative therapies and approaches.

Targeting of the alveolar epithelium may be an alternative therapeutic approach but there are currently a lack of models with which to study potential early changes in the epithelium, a stage at which pharmacological treatment could have the most impact on preventing rapid disease progression. We believe our model can help fill that gap.

2. The results are all obtained with one cell line and effects can differ markedly between cell lines. Can the authors confirm using a second hiPSC line that main effects of FC can be reproduced?

R2: We thank the reviewer for this suggestion.

We agree with the reviewer that the reproducibility and variability in response to the FC is an important aspect to consider depending on the application. The aspect of reproducibility of the effects of the FC was discussed in the original report describing the FC stimulation in human precision-cut-lung slices (Alsafadi et al. 2017). We found that while the effects of the FC were generally consistent across donors, some patients donors responded differently for yet unknown reasons. We therefore focused on the reproducibility of the FC within one starting iPSC line as this is the most likely scenario for a primary drug screen. Due to the challenges associated with high-throughput drug screens searching for novel compounds, they are almost always performed using one cell line. This is then followed by secondary screening in additional patients or additional models with positive hits identified in the primary drug screen.

Therefore, repeating the full differentiation/ cryopreservation/ expansion protocol to assess whether the FC can work on an additional cell line is an important question to address, we feel it is beyond the scope and requirements of this particular study. We would like to highlight that only using one additional cell line would not be sufficient to answer the broader question as to how applicable is the FC to wider patient populations and it would likely require more than just 2 or 3 additional iPSC lines to be assessed properly. Future work by us and other labs should of course investigate this and further characterise this with more cell lines to understand its potential for disease modeling of diverse patients or in personalised medicine approaches.

We have included additional Discussion that hopefully covers our rationale above.

Alsafadi et al. 2017: <https://pubmed.ncbi.nlm.nih.gov/28314802/>

3. The choice of testing Nintedanib and Pirfenidone in this alveolar type-2 only model is a bit puzzling to me, especially with regard to pirfenidone. Both Nintedanib and Pirfenidone target predominantly the fibroblasts compartment, with Nintedanib additionally affecting VEGFR. Nintedanib could have therefore impacted other cell-types such as endothelial cells, for example impacting angiogenesis, but their effects on epithelial (type-2) cells should be limited.

R3: Thank you for raising this important point to clarify. Previous work by some of the authors of this paper (see Lehmann et al. 2018) and others have described significant anti-fibrotic effects of Nintedanib and Pirfenidone in primary mouse alveolar epithelial type 2 cells (increase in *Sftpc*, alveolar type 2 marker and reduction of *Fn1*, extracellular matrix component). The observations from using Nintedanib and Pirfenidone in our human alveolar epithelial system thus complements this published study by showing that the cells in our system are responsive to treatment in some aspects similarly to the mouse alveolar epithelial type 2 cells.

Lehmann et al. 2018: <https://pubmed.ncbi.nlm.nih.gov/30219058/>

We would also like to point out that at the time these two drugs were approved and entered clinical use, there was very limited knowledge around the aberrant epithelial phenotypes present in the IPF lung. While epithelial biomarkers were found to be aberrant in the PROFILE cohort (Maher et al. 2017), the use of Pirfenidone and Nintedanib was limited at that time to a small subset of patients and was thus underpowered. Further, there was no possibility in these clinical trials to assess if these drugs had an impact on aberrant cell populations such as the aberrant basaloid cells, which were only described in single-cell sequencing studies of the IPF lung published in 2017 and onwards (Xu et al. 2016, Reyfman et al. 2019, Adams et al. 2020, Habermann et al. 2020). This is an aspect we have been able to demonstrate through the use of our model as described in this study.

Maher et al. 2017: [https://www.thelancet.com/journals/lanres/article/PIIS2213-2600\(17\)30430-7/fulltext](https://www.thelancet.com/journals/lanres/article/PIIS2213-2600(17)30430-7/fulltext)

Xu et al. 2016: <https://pubmed.ncbi.nlm.nih.gov/27942595/>

Reyfman et al. 2019: <https://pubmed.ncbi.nlm.nih.gov/30554520/>

Adams et al. 2020: <https://pubmed.ncbi.nlm.nih.gov/32832599/>

Habermann et al. 2020: <https://pubmed.ncbi.nlm.nih.gov/32832598/>

4. Can the authors provide some clarity about the purity of the organoids? Can the authors for example stain the organoids with KRT5? I am a bit surprised to see a baseline gene expression level of KRT5 in Fig. 3C and wonder what the purity is of the total organoid population in culture. Are all organoids expressing AT2 markers or is a percentage of organoids KRT5-positive as they perhaps went into the direction of airway? KRT17 is also expressed by basal cells and the staining of the CC cultures in Fig. 3D also suggest that within an organoid there may be some basal cells present?

R4: Thank you for this suggestion. We have provided this additional analysis in Supplemental Figure 3. In our organoid population under non-injurious conditions, we have around 74% of organoids expressing the AEC2 marker pro-SP-C and we could not detect any organoids positive for KRT5 on protein level.

Similarly to what we observed in our study, transcript levels of *KRT5* are detected in primary AEC2 organoids also in other studies (please see Figure 4B in Kathiriya et al. 2022). In the same study, no KRT5+ cells could be detected by immunostaining of iPSC-derived AEC2 organoids cultured in this same format (please see Figure 1H and I in Kathiriya et al. 2022). It is therefore possible that the same applies to our iAEC2 cultures, i.e. that the low baseline level of *KRT5* transcripts detected by sequencing does not give rise to KRT5+ cells detectable by immunostaining.

Kathiriya et al. 2022: <https://pubmed.ncbi.nlm.nih.gov/34969962/>

KRT17 transcript and protein expression has also been recently observed by other groups in their iAEC2 3D cultures (including without the use of a stromal support cell (Alysandratos et al. 2023). Possibly, the low expression level of some basal cell-associated transcripts such as KRT17 in non-fibrotic alveolospheres could be due to a subportion of cells being less mature in their AEC2-like phenotype (akin to transitional cell types recently described in other human lung organoid papers, please see Alysandratos et al. 2023 and Kathiriya et al. 2022).

Alysandratos et al. 2023 <https://pubmed.ncbi.nlm.nih.gov/36454643/>

5. Figure 5 indicates polarized collagen I expression according to the authors, but the organoid that is depicted in the image has no lumen, a by the authors called: 'dense' organoid. Can there still be secretion from any other location than basal in that case? So is this indeed polarization or a normal consequence of a missing lumen?

R5: We thank the reviewer for this suggestion. In the revised Figure 6, we have replaced the immunofluorescence images as suggested by another reviewer with images from non-permeabilised staining conditions which confirm the secretion of COL1 on the basal side in both “normal” and “dense” organoids. We have moved the images of permeabilised organoids to Supplemental Figure 10 and have additionally included images of both “normal” and “dense” organoids in that panel. These images indicate that the secretion of collagen 1 is occurring on the basal side of the FC-stimulated organoids, even in organoids where secretion into the lumen would be physically possible.

6. Page 7 line 140: was there also a reduction in the number of organoids? Was this assessed? And how was the incubation of 72 h chosen?

R6: We thank the reviewer for this comment. The number of organoids detected for each of the described experiments in the manuscript has now been added in Supplemental Figures 4D, 11C and 13C. We could not see a consistent reduction of the number of organoids with the FC stimulation.

Our rationale for the FC stimulation of 72 h of the alveolospheres was to avoid medium changes during the stimulation to be able to measure the total concentration of secreted mediators. We also hypothesised that the FC conditioned medium, containing both the added factors and the secreted factors from the injured cells, could potentially propagate the effects of the FC. 72 h is the maximum amount of time between medium changes during maintenance culture of the alveolospheres without impact on their growth and so this guided our selection for the stimulation. Our chosen time point of FC stimulation is well in line with the reported findings in the original paper describing the FC stimulation in human precision-cut lung slices (Alsafadi et al. 2017) where a significant reduction of surfactant protein C is seen on protein level already at 48 h and further on transcriptomic and protein level at 120 h (our time point of 72 h is between the 48 h and 120 h).

Alsafadi et al. 2017 <https://pubmed.ncbi.nlm.nih.gov/28314802/>

7. The authors report no visible changes after FC withdrawal (p15, line 355-357), but the images in Fig. 6G suggest a much larger lumen or is this not a good representation in that respect?

R7: Thank you for posing this question. We have now included quantitative measurements of both size and changes in density of the organoids in Figure 7C and D in the revised manuscript to aid the interpretation of the morphology, showing that neither the average size nor density of the organoids is reverted after the withdrawal of the FC. To visually support these observations better, we have included brightfield images of FC stimulated alveolospheres. We thank the reviewer for this comment as the addition of the suggested new data provides clearer support to our statement about the preserved dense morphology of the organoids after the FC withdrawal.

April 5, 2023

RE: Life Science Alliance Manuscript #LSA-2022-01853-TR

Dr. Lynne Anne Murray
MiroBio
Winchester House
Oxford Science Park
Oxford, Oxfordshire OX4 4GE
United Kingdom

Dear Dr. Murray,

Thank you for submitting your revised manuscript entitled "Modeling Fibrotic Alveolar Transitional Cells with Pluripotent Stem Cell-derived Alveolar Organoids". We would be happy to publish your paper in Life Science Alliance pending final revisions necessary to meet our formatting guidelines.

- please add the Twitter handle of your host institute/organization as well as your own or/and one of the authors in our system
- please use the [10 author names, et al.] format in your references (i.e. limit the author names to the first 10)
- please add the supplementary figure legends to the main manuscript text

A. FINAL FILES:

B. MANUSCRIPT ORGANIZATION AND FORMATTING:

**Submission of a paper that does not conform to Life Science Alliance guidelines will delay the acceptance of your

manuscript.**

The license to publish form must be signed before your manuscript can be sent to production. A link to the electronic license to publish form will be sent to the corresponding author only. Please take a moment to check your funder requirements.

Sincerely,

Reviewer #1 (Comments to the Authors (Required)):

Dear Dr. Guidi,
Ptasinski et al., have made significant changes to the manuscript after revision. They have addressed the concerns raised in the first revision. Importantly, they have also adequately discussed and addressed the shortcomings of the study fairly. I would recommend the manuscript to be published as such now in LSA.

As for the question, Do you consider the conclusions of the paper justified based on the presented data? (Please briefly explain in your report)

Yes I consider the paper justifies the conclusion based on the presented data.

Thanks
best wishes

Reviewer #2 (Comments to the Authors (Required)):

The authors addressed my concerns.

Reviewer #3 (Comments to the Authors (Required)):

Thank you for the clear answers to the items I addressed. Very nice work.

May 10, 2023

RE: Life Science Alliance Manuscript #LSA-2022-01853-TRR

Dr. Lynne Anne Murray
MiroBio
Winchester House
Oxford Science Park
Oxford, Oxfordshire OX4 4GE
United Kingdom

Dear Dr. Murray,

Thank you for submitting your Research Article entitled "Modeling Fibrotic Alveolar Transitional Cells with Pluripotent Stem Cell-derived Alveolar Organoids". It is a pleasure to let you know that your manuscript is now accepted for publication in Life Science Alliance. Congratulations on this interesting work.

DISTRIBUTION OF MATERIALS:

Again, congratulations on a very nice paper. I hope you found the review process to be constructive and are pleased with how the manuscript was handled editorially. We look forward to future exciting submissions from your lab.

Sincerely,
